# Identification of NLOS and Multi-Path Conditions in UWB Localization Using Machine Learning Methods

**Cung Lian Sang \***[ID]**, Bastian Steinhagen**[ID]**, Jonas Dominik Homburg**[ID]**, Michael Adams**[ID]**, Marc Hesse**[ID] **and Ulrich Rückert**

Cognitronics and Sensor Systems Group, CITEC, Bielefeld University, 33619 Bielefeld, Germany;
bsteinhagen@techfak.uni-bielefeld.de (B.S.); jhomburg@techfak.uni-bielefeld.de (J.D.H.);
madams@techfak.uni-bielefeld.de (M.A.); mhesse@techfak.uni-bielefeld.de (M.H.);
rueckert@techfak.uni-bielefeld.de (U.R.)
\* Correspondence: csang@techfak.uni-bielefeld.de; Tel.: +49-521-106-67368

**Abstract:** In ultra-wideband (UWB)-based wireless ranging or distance measurement, differentiation between line-of-sight (LOS), non-line-of-sight (NLOS), and multi-path (MP) conditions is important for precise indoor localization. This is because the accuracy of the reported measured distance in UWB ranging systems is directly affected by the measurement conditions (LOS, NLOS, or MP). However, the major contributions in the literature only address the binary classification between LOS and NLOS in UWB ranging systems. The MP condition is usually ignored. In fact, the MP condition also has a significant impact on the ranging errors of the UWB compared to the direct LOS measurement results. However, the magnitudes of the error contained in MP conditions are generally lower than completely blocked NLOS scenarios. This paper addresses machine learning techniques for identification of the three mentioned classes (LOS, NLOS, and MP) in the UWB indoor localization system using an experimental dataset. The dataset was collected in different conditions in different scenarios in indoor environments. Using the collected real measurement data, we compared three machine learning (ML) classifiers, i.e., support vector machine (SVM), random forest (RF) based on an ensemble learning method, and multilayer perceptron (MLP) based on a deep artificial neural network, in terms of their performance. The results showed that applying ML methods in UWB ranging systems was effective in the identification of the above-three mentioned classes. Specifically, the overall accuracy reached up to 91.9% in the best-case scenario and 72.9% in the worst-case scenario. Regarding the F1-score, it was 0.92 in the best-case and 0.69 in the worst-case scenario. For reproducible results and further exploration, we provide the publicly accessible experimental research data discussed in this paper at PUB (Publications at Bielefeld University). The evaluations of the three classifiers are conducted using the open-source Python machine learning library scikit-learn.

**Keywords:** UWB; NLOS identification; multi-path detection; NLOS and MP discrimination; machine learning; SVM; random forest; multilayer perceptron; LOS; DWM1000; indoor localization

## 1. Introduction

Indoor localization systems enable several potential applications in diverse fields. A few examples where positioning is crucial include tracking valuable assets and personal devices in IoT, ambient assisted living systems in smart homes and hospitals, logistics, autonomous driving systems, customer tracking systems in shopping and public areas, positioning systems in industrial environments, and mission-critical systems such as an application for firefighters and soldiers [1–3]. Among several technologies available for indoor localization described in the literature, ultra-wideband (UWB)

technology [1,4,5] plays an increasingly important role in precise indoor localization systems due to its fine ranging resolution and obstacle-penetration capabilities [2,3,6].

In wireless ranging systems including UWB technology, the distance between the transmitter and receiver is estimated by measuring the time-of-flight (TOF) between the two transceivers and multiplying it by the speed of light [7,8]. However, the ranging algorithm assumes that the *TOF* signal is always in a direct line-of-sight (LOS) condition. Therefore, non-line-of-sight (NLOS) [9–11] and multi-path (MP) [3] conditions cause a positive bias in the estimated distances. Figure 1 expresses an abstract view of the LOS, NLOS, and MP conditions in typical wireless communications. The figure shows how a signal sent from a tag device (green pyramid shape in the middle) can be received in different scenarios at the anchor nodes (yellow pyramid shapes). In Figure 1, we define two possible MP conditions in wireless communication. The first condition is clear because the first path signal is completely blocked by the obstacle, and the only received signal in the measurement is based on the bounded signal from the transmitter. However, distance measurement in wireless communication could also be distorted by multiple reflections of signals even if there is no direct obstacle between the transceivers. For instance, wireless measurement is conducted in a narrow corridor, tunnel, etc. We confirmed the error caused by such MP conditions using UWB in our previous work [8]. The research data of the mentioned work are publicly available in [12]. Similar results based on UWB were also reported in [13]. Therefore, differentiation between the LOS, NLOS, and MP conditions in wireless ranging systems is important for precise localization systems.

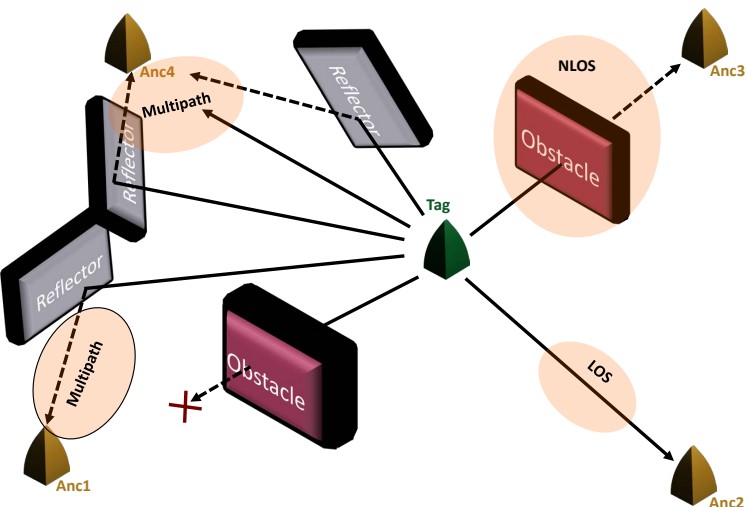

**Figure 1.** Illustration of LOS, NLOS and multi-path (MP) scenarios in a UWB-based ranging system.

This paper discusses the vital role of classifying the LOS, NLOS, and MP scenarios in UWB ranging system using machine learning (ML) approaches. By understanding the defined three classes, a positioning algorithm [6,14] can mitigate the biases caused by NLOS and MP conditions, i.e., by giving different weights to each class. We proposed such a mitigation technique in our previous work [2]. The common identification and mitigation techniques for the NLOS condition in UWB can be found in [9,15,16] and the references provided therein.

In fact, the multi-class identification of UWB measurement data (LOS, NLOS, and MP) in the real world is challenging in indoor environments because a variety of physical effects can distort the direct path LOS signal in different ways [3,17] (Figure 1). This includes walls, furniture, humans, the orientation of the UWB antenna, etc. Therefore, machine learning methods are attractive solutions for solving such a problem.

Identification and mitigation techniques of the NLOS condition in UWB, or wireless communications in general, using ML methods are not new. It has received significant interest

recently years [9–11,17–23]. However, the major contributions in the literature address the binary classification between the LOS and NLOS in the UWB ranging system.

In contrast, this paper addresses machine learning techniques for direct identification of the three mentioned classes (LOS, NLOS, and MP) in a UWB indoor localization system using experimental data collected in seven different environments in two different test scenarios (Section 4). Using the collected real measurement data, we compare three machine learning methods, i.e., support vector machine (SVM), random forest (RF), and multilayer perceptron (MLP), in terms of their prediction accuracy, training time, and testing time. The classifiers are chosen by bearing in mind that the evaluated ML models can be used in a low cost and power efficient real-time system such as microcontroller-based platforms [24].

For the sake of reproducible results and further exploration, we provide all the experimental research data and the corresponding source codes as the Supplementary Data of this manuscript in PUB (Publications at Bielefeld University) [25], which is publicly available. The evaluation of the algorithms is conducted using the open-source Python machine learning library scikit-learn [26].

## 2. Problem Description

The primary goal of the identification process in wireless communications is to detect the existence of an NLOS and/or MP condition in a communication between a transmitter and a receiver. This process is crucial because the multi-path effects and the NLOS conditions strongly influence the accuracy of the measured distances in wireless communications. As an example, Figure 2d compares the error of measured distances in static scenarios in the LOS, NLOS, and MP conditions in UWB based on our experimental data. Figure 2a–c also illustrates the comparison of the conventional identification techniques for the three mentioned classes (LOS, NLOS, and MP) based on the first-path (FP) power level and the channel impulse response (CIR) (a more detailed description is given in Section 3.1).

The experimental evaluation results in Figure 2d suggest that the magnitude of the error in NLOS and MP conditions is considerably larger than the LOS condition compared to the ground truth reference. Moreover, the error introduced by the MP condition is significantly lower than the completely blocked NLOS condition, where the signal needs to penetrate the obstacle to reach the receiver. Indeed, this depends on the materials and other factors of the obstacles [3]. However, the result in Figure 2d, which is blocked by a human in this experiment, indicates that the NLOS condition introduces the highest impact on the measured distance errors. This motivated us to classify the UWB ranging systems into three classes (LOS, NLOS, MP) to improve the location accuracy in the UWB localization system. The classified ranging information is applicable in any positioning algorithm [6,14] to mitigate the biases effectively [2,18,20] caused by the NLOS and MP conditions. It should be noted that the measured distances in Figure 2d were conducted in the static scenario at approximately a 6 m distance between the anchor and tag for the three classes. The ground truth references of the distance were measured using a laser distance meter, CEM iLDM-150 (http://www.cem-instruments.in/product.php?pname=iLDM-150), which provided an accuracy of ±1.5 mm according to the datasheet of the manufacturer. Regarding the ranging errors in UWB in a static scenario, a more rigorous evaluation was conducted in our previous work [8], where its corresponding experimental research data were given in [12]. The result of MP conditions in Figure 2d was based on a scenario when there was no obstacle between the transceivers, as illustrated in Figure 1 (Section 1). To be precise, the measurement was conducted in indoor environments where multiple reflections from walls occurred in a narrow corridor. One of the reasons the error occurred in the MP condition in this scenario was because of the preamble accumulation time delay (PATD) in the coherent receiver of the UWB chip [8]. PATD is affected by the presence of multi-path in wireless measurements. It is more notable when the arrivals of the reflected signal are within the chip period of the first path signal [8]. Moreover, it is worth mentioning that the error deviations in Figure 2d correspond to a single range (a tag to an anchor) in the measurement. In general, at least three ranges (usually more in multilateration methods) of such

measures are necessary for UWB localization [6,14]. This implies that the combination of such errors in a ranging phase contributes to a significant impact on the overall system performance.

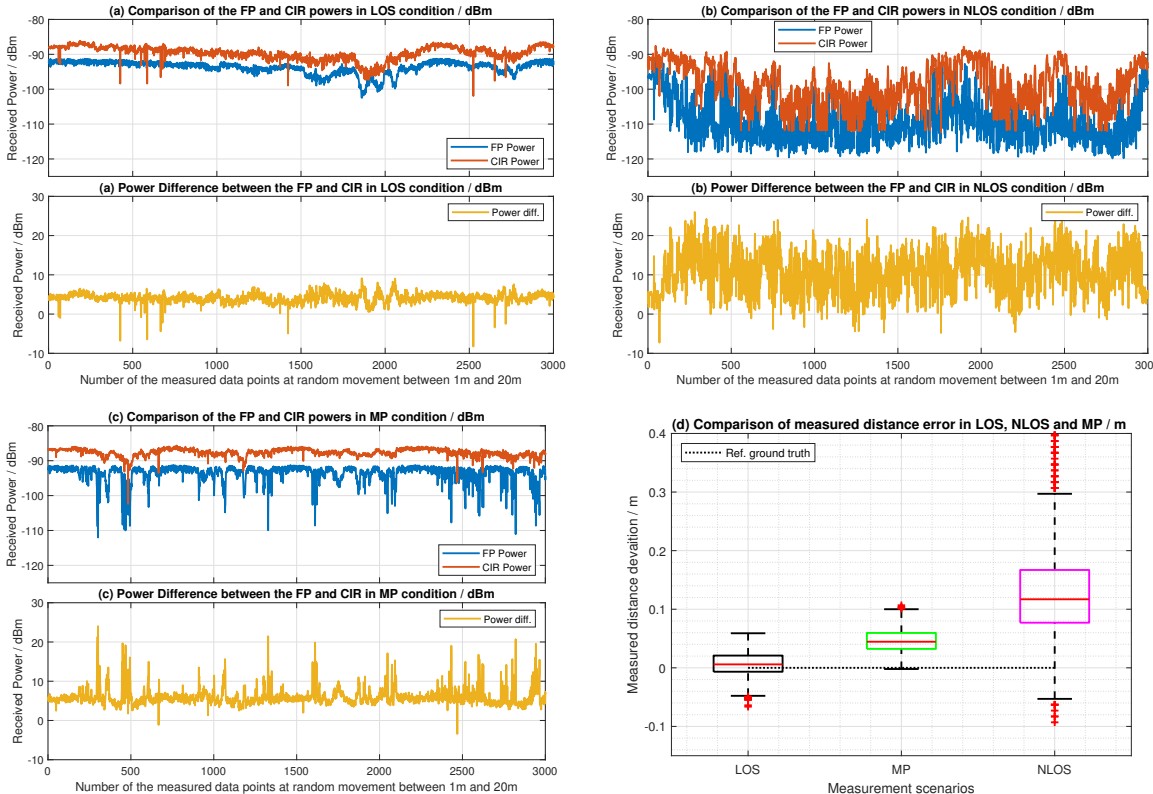

**Figure 2.** Illustration of the LOS, NLOS, and MP conditions in UWB ranging systems: (**a**–**c**) Comparison of the FP power, the CIR power, and the difference between the two power levels (FP and CIR power) in three scenarios: (**a**) LOS, (**b**) NLOS, and (**c**) MP conditions. The measurement was conducted for the three scenarios for random movement between 1 m and 24 m distances. (**d**) Comparison of the measured distance errors in the three mentioned conditions (LOS, NLOS, and MP) in static scenarios.

## 3. Related Works

In the literature, the problem solving strategy for ranging errors in UWB due to the effects of NLOS and/or MP can be coarsely classified into two steps [9,15]: (i) the NLOS identification process [16,27,28] and (ii) the NLOS mitigation process [9,16,29]. This paper is solely focused on the former case. In fact, there exists a method that bypasses the identification process and directly mitigates the ranging error using channel statistics and SVM as an ML-based classifier [18,30]. However, this method restricts the flexibility to choose different positioning algorithms in the latter case since the mitigation technique is limited to a few compatible algorithms.

The common approach is to detect the non-direct path signal (i.e., NLOS and/or MP) and use the detected information to modify the location algorithm in order to mitigate the biases caused by the NLOS and/or MP conditions [9,10,17,18,20,21,30]. In this manuscript, we divided the related works into two subsections: (i) conventional approaches without using ML techniques (Section 3.1) and (ii) ML-based approaches (Section 3.2). Our proposed technique regarding the multi-classes identification process in UWB was based on the ML-based approach.

### 3.1. Conventional NLOS Identification Techniques in UWB

As already mentioned in Section 1, identification of NLOS and LOS in UWB communications is not new. There have been several proposals in the literature [9,16,27,28,31–33] to identify and mitigate the NLOS conditions in UWB. However, the identification process of the MP condition is usually

ignored in the literature, although the effects of the MP conditions in UWB ranging systems were acknowledged as important aspects in [3,8,13,21]. Conventionally, the NLOS detection in UWB is always regarded as a binary classification problem. The traditional NLOS detection methods can be coarsely categorized as:

- Identification of the NLOS situation based on a binary hypothesis test [27]
- NLOS detection based on the change of the signal-to-noise ratio (SNR) [28]
- NLOS identification based on channel impulse response [9,31]
- NLOS detection techniques based on the multi-path channel statistics such as the kurtosis, the mean excess delay spread, and the root mean squared delay spread [16,33]
- Detection of the NLOS condition using the received signal strength (RSS) [32,33]

In brief, the conventional NLOS identification approaches mainly rely on the statistical condition of the received signals in UWB communications. Figure 2a–c demonstrates these scenarios by comparing the first-path (FP) signal power and the channel impulse response (CIR) power for the three conditions (LOS, NLOS, and MP). Among the mentioned NLOS detection methods, the threshold approach presented by Decawave in [34] has been widely used in different UWB applications and system implementations [23,34,35]. This is accomplished by taking the difference between the estimated total received (RX) power and the first-path (FP) power using the following equations [34] (Figure 2a–c):

$$FP\ Power\ Level = 10 \cdot log_{10}(\frac{F_1^2 + F_2^2 + F_3^2}{N^2}) - A \tag{1}$$

where $F_1$, $F_2$, and $F_3$ are the first, second, and third harmonics of the first-path signal amplitudes for signal propagation through wireless media as in the multi-path, NLOS, and/or LOS scenarios [34,35]. $N$ is the value of the preamble accumulation count reported in the DW1000 chip from Decawave. $A$ is a predefined constant value, which has 133.77 for a pulse repetition frequency (PRF) of 16 MHz and 121.74 for a PRF of 64 MHz.

The estimated received power (RX) level can be defined as:

$$RX\ Power\ Level = 10 \cdot log_{10}(\frac{C \cdot 2^{17}}{N^2}) - A \tag{2}$$

where $C$ is the value of the channel impulse response power reported in the DW1000 chip.

Therefore, the metric that specifies the conditions of LOS and NLOS in the threshold method can be achieved by computing the difference between the received and first-path power [34] as:

$$Threshold\ Power = RX\ Power\ Level - FP\ Power\ Level \tag{3}$$

In the conventional threshold approach, the measured distance is classified as a LOS when the threshold power using (3) is less than 6 dBm and defined as an NLOS when it is more than 10 dBm [34]. This is a sub-optimal acceptable solution as our particular experimental evaluation results show in Figure 2. That is, the mean value of the threshold power including its standard deviation in the LOS condition using (3) is $4.12 \pm 1.13$ dBm (Figure 2a) and in the the NLOS condition is $10.75 \pm 5.51$ dBm (Figure 2b). However, the solution is not optimal as much fluctuation can occur as described in the experimental measurement data (Figure 2a–c). The condition is harsher to solve in MP condition, where the first-path signals are hard to distinguish clearly from the received signal (Figure 2c).

Nevertheless, the classification of the three mentioned classes is not straightforward. The complexity of the classification problem increases especially in indoor environments because of several factors such as material characteristics [36], the refractive index of different materials, and so on [1,3]. Moreover, the phenomenon of the multi-path effects and NLOS depends on the properties of the medium through which the signal travels, the location (dimension of the places and rooms)

where the signals are measured, the presence of other objects within the measured environment, the orientation of the UWB antenna, etc. Therefore, ML approaches have been regarded as attractive strategies for solving this complex task in recent years (Section 3.2).

### 3.2. Identification of the NLOS and MP Conditions in the Literature Based on Machine Learning Techniques

One of the earlier ML-based NLOS identifications in UWB was conducted in [9] using SVM as a classifier. In that paper, the identification process was considered as a binary classification problem (LOS vs. NLOS) showing that the ML approaches outperformed the traditional parametric techniques and signal processing approaches from the literature.

Consequently, several investigations of the NLOS identification process in UWB were examined in the literature using different ML techniques as a classifier such as SVM in [9,10,17,21], MLP in [23,37,38], boosted decision tree (BDT) in [38], recursive decision tree in [35], and other ML techniques such as kernel principal component analysis in [19], etc. Moreover, the unsupervised machine learning technique called "expectation maximization for Gaussian mixture models" was recently applied in [39] to classify the LOS and NLOS conditions in UWB measurement. Likewise, deep learning approaches such as the convolutional neural network (CNN) were also explored to distinguish the NLOS condition from LOS in UWB ranging [13,22]. In CNN-based deep learning approaches, the authors generally modified the existing CNN network such as GoogLeNet [22], VGG-architecture (i.e., VGG-16 or VGG-19) [13,22], AlexNet [22], etc., to be usable for the low cost UWB systems. The reported overall accuracy ranges started from 60% (using a typical ML technique such as SVM) up to 99% (using the CNN approach). In all of the above-mentioned approaches, the focus is solely on detecting the NLOS condition in UWRranging, i.e., the binary classification between LOS and NLOS.

Moreover, the performance comparison of different ML techniques for the identification of NLOS in UWB ranging was conducted in [11,17,38]. The main purpose of these analyses was to compare the impact of model selection in ML-based system applications in UWB. In [38], the performance comparison of two ML methods namely MLP and BDT was carried out for the binary classification (LOS vs. NLOS). The resultant report concluded that the BDT outperformed the MLP. Likewise, the comparison of five classifiers using MATLAB (i.e., SVM, k-nearest neighbor (KNN), binary decision tree, Gaussian process (GP), generalized linear model) was performed in [11]. The authors concluded that KNN and GP performed better than the other three models. Similarly, the authors in [17] evaluated three ML models (SVM, MLP, RF) to classify the LOS and NLOS in narrowband wireless communications (i.e., not specifically for UWB systems in this case). The authors reported that RF and MLP performed better than SVM in all of their evaluations.

In contrast to the binary classification between LOS and NLOS in UWB, the binary classification of the MP from LOS conditions was investigated in [13]. The author reported that MP effects could cause an error in UWB ranging from a few centimeters up to 60 cm. Similar deviation of the error in the MP condition can be seen in our experimental evaluation presented in Figure 2d.

Throughout the literature, the problem has been treated as a binary classification problem or hypothesis test (i.e., LOS vs. NLOS or LOS vs. MP). To the best of the authors' knowledge, only two papers addressed UWB-based ranging errors as a multi-class problem [11,21]. The first paper was based on a two-step identification process [21] using SVM as a classifier. In that paper, the LOS and NLOS were identified in the first step. Then, further classification (MP vs. NLOS) was categorized in the second step if NLOS was detected in the first step. The second paper categorized the NLOS conditions into two types (soft-NLOS vs. hard-NLOS) in addition to LOS while ignoring the MP effects [11]. The differentiation between the two NLOS types was primarily based on the material of the obstacles, which the UWB signal was passing through by penetration. The authors used two types of walls in their evaluation to classify a soft-NLOS and a hard-NLOS.

In contrast with the above-mentioned approaches, we performed a direct identification of the multi-class classification for UWB ranging systems in this paper. The classified classes were the LOS, NLOS, and MP conditions. Based on the measurement data, we performed three ML models (Section 5),

namely SVM, RF, and MLP, to compare their performances (Sections 6 and 7). The experimental research data utilized in this paper [25] are provided on a public archive for reproducible results and further exploration.

## 4. Measurement Scenarios and Data Preparation

In this section, we describe the experimental setup of the evaluations (Section 4.1), the data collection processes including labeling and data separation (Section 4.2), and the feature extraction based on the collected data (Section 4.3) for the three evaluated ML models.

### 4.1. Experimental Setup

For the UWB data measurement process in the experimental evaluations, we used a DWM1000 module [34] manufactured by Decawave as the UWB hardware and the STM32 development board (NUCLEO-L476RG) [40] manufactured by STMicroelectronics as the main microcontroller (MCU). Table 1 provides the hardware configurations used in the experimental evaluations.

**Table 1.** Configurations of the primary hardware used in the experimental evaluation.

| Types of Hardware | Properties | Values |
|---|---|---|
| UWB module | Module name | DWM1000 |
| | Data rate | 6.8 Mbps |
| | Center frequency | 3993.6 MHz |
| | Bandwidth | 499.2 MHz |
| | Channel | 2 |
| | Pulse-repetition frequency (PRF) | 16 MHz |
| | Reported precision | 10 cm |
| | manufacturer | Decawave |
| Microcontroller (MCU) | Module type | STM32L476RG |
| | Development board | NUCLEO-L476RG |
| | Manufacturer | STMicroelectronics |

Our previous work [7,8] pointed out that the alternative double-sided two-way ranging (AltDS-TWR) method outperformed other available TWR methods in the literature in different tested scenarios. Therefore, we applied AltDS-TWR as a wireless ranging method in our evaluations. Furthermore, AltDS-TWR operated well without the need to use high precision external oscillators in the MCU [8]. Hence, the built-in high-speed internal (HSI) clock source (16 MHz) from the MCU was applied to all of the evaluation results presented in this article. According to the data sheet [41], the HSI has an accuracy of $\pm 1\%$ using the factory-trimmed RC oscillator.

During measurement, one of the transceivers among the two was connected to a computer for logging the data via a serial USART port. Both transceivers were executed with a two-way ranging software provided by Decawave for production testing of their evaluation kit (EVK1000), which is available on Decawave's website (https://www.decawave.com/software/). We modified the library of this software to extract all required features provided in Section 4.3. Then, we logged and saved the extracted features into a file for each trial in our measurement campaign. To avoid the effect of Fresnel zones in our measurement results, the antenna height was always maintained at 1.06 m in one of our UWB devices, i.e., the static one that recorded the measurement data via PC.

### 4.2. Data Collection Process

The required data for experimental evaluations presented in Section 7 were collected in three scenarios (two small rooms, a hall, and four corridors) at seven different places in indoor environments (Figure 3). The two rooms were the (6 m × 6 m) laboratory environment and approximately the (8 m × 6 m) communication room in which different items of furniture were placed. The collected data in narrow corridors were intended for MP conditions, where the direct LOS could

not be distinguished because of multiple signal reflections from the narrow walls. Figure 2c illustrates a concrete example of this MP condition in terms of the FP and RX powers. In all cases, the data were collected for both static and dynamic conditions. In the dynamic case, the device attached to the PC stayed static, while another device was held by a human during random walks. In the static scenario, the two transceivers were at a vertical position of 90° pointing to the antenna of the DWM1000 module as an upward position without any rotation. However, the antenna of the device held by a human during the dynamic condition was randomly rotated between 0° and 180° in some cases of the data collection process. Moreover, the NLOS conditions by blocking the communication between two transceivers using a human as an obstacle were conducted in all cases depicted in Figure 3. Besides, a thick concrete wall, pieces of concrete block, and a mixture of wood and metal were also applied as parts of the obstacles for NLOS conditions in the two small rooms and their environments.

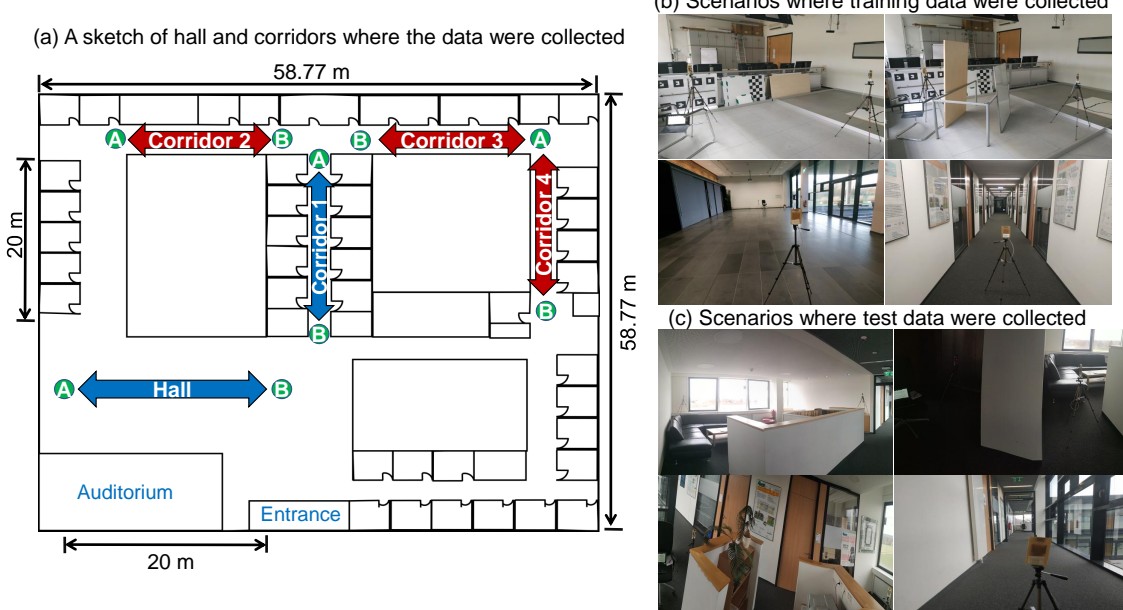

**Figure 3.** Illustration of the scenarios where training and test data were collected for evaluation: (**a**) A sketch of the building where the experimental data for training and test were collected. (**b**) Training data were collected for LOS, NLOS (including human blocking) and MP conditions in a laboratory, a large hall, and a corridor (blue color in (**a**)). (**c**) Similar to (**b**), test data were collected in a different room (including different types of furniture and NLOS human blocking) and a different corridor (red in (**a**)).

### 4.2.1. Labeling the Measured Data and Dealing with the Class Imbalance Case

The class labels (LOS, NLOS, and MP) were manually annotated in the data preprocessing phase after the measurement campaign. During the data measurement process, a block of observations for each trial regarding the three categories (LOS, NLOS, MP) was carried out and saved individually into the PC. The block referred to a collection of data belonging to the same class that was saved separately in the computer as a single file. For instance, we collected the random movement data in Corridor 2 (Figure 3) for 5 min using a data update rate of the system as 20 Hz. At the same time, we made sure that there were no obstacles between the two transceivers, and the antenna was held still at 90° during the process. Then, we annotated the block of this whole measured data as an MP class to be used in our evaluation. In this way, the whole block of that data in each evaluated scenario could easily be labeled as an LOS, NLOS, or MP.

The initial UWB dataset achieved from the measurement was imbalanced for the three demanded classes (LOS, NLOS, and MP), which is a typical phenomenon in data collection. Class imbalance refers to a scenario where the number of observations in each class is not the same in the measurement.

In other words, the number of samples in one class or more classes is significantly lower or higher than those belonging to the other classes. There are several techniques to deal with the imbalanced data in classification problems including resampling techniques and algorithms [42].

We chose a random undersampling technique [42] in our evaluation to balance the three mentioned classes equally. This ensured that no artificial data points were created outside of the measured experimental data. The undersampling was performed by setting the class belonging to the smallest number of observations as a base class. Then, the classes belonging to higher samples of observation were reduced to balance with the total number of the base class by randomly selecting their elements.

### 4.2.2. Separation of the Training, Validation, and Test Dataset

Two independent test datasets were used in our experimental evaluations. The first test dataset was separated from the measurement environments provided in Figure 3b, i.e., the environments from which the training data came. This is the typical scenario for the presentation of classification results in most UWB-based literature [10,11,21]. In some cases in the literature, the test dataset was separately collected by intentionally switching the subject of the experiment, i.e., a person who carried the UWB device in [11]. However, the environment of the measurement stayed unchanged in the evaluation. As already mentioned in Section 2, the propagation of the measured UWB signal can be affected by several environmental factors such as the refractive index of materials, the placement of objects in the measured circumstances, etc. To examine this incident in our results, we collected a second test dataset that was independent and different from the training environments (Figure 3c). This second dataset was solely split out for the purpose of testing in our evaluation. The results using both test datasets are discussed in Section 7.

For training the evaluated ML models including the validation process, more data were gathered in the laboratory room, the hall, and the first corridor (Figure 3b). In particular, one-hundred eighty-five-thousand seven-hundred ninety observations in total were gathered after balancing the three classes in this scenario. This meant each class belonged to 61,930 data points. For testing purposes, thirty percent of the data points were left out by random shuffling in each trial conducted in Section 7. The results using this test scenario in Figure 3b are expressed as a scenario when test and training are in the same condition (Section 7).

In contrast, the measurement campaign, particularly for testing purposes, was conducted in different scenarios from the training. These measurements were carried out in a different room with various items of furniture and three different corridors (Figure 3c). The results achieved from these second test scenarios are expressed in Section 7 as a scenario where the test and training conditions were different. The total number of 36,015 data points, 12,005 for each class, was used for conducting this test scenario in our evaluation after balancing equally the three classes.

### 4.3. Feature Extraction

In total, twelve features were extracted from the DWM1000 UWB modules manufactured by Decawave [34] using the configuration described in Table 1. The extracted features were based on the typical parameters that are necessary in the traditional NLOS identification methods as expressed in Equations (1)–(3). This meant that no extra burden was involved by using these extracted features in our ML application. For the sake of completeness, two more features namely standard and maximum noises supported by the DW1000 module were included in the feature extraction of our evaluation. Therefore, the full features extracted and saved during the experimental evaluation were:

1. the reported measured distance
2. the compound amplitudes of multiple harmonics in the FP signal
3. the amplitude of the first harmonic in the FP signal
4. the amplitude of the second harmonic in the FP signal

5.  the amplitude of the third harmonic in the FP signal
6.  the amplitude of the channel impulse response (CIR)
7.  the preamble accumulation count reported in the DW1000 chip module
8.  the estimated FP power level using (1)
9.  the estimated RX power level using (2)
10. the difference between the FP and RX power level using (3)
11. the standard noise reported in the DW1000 chip module
12. the maximum noise reported in the DW1000 chip module

Regarding the above-mentioned 12 features, we would like to mention that the feature extraction was solely based on the DW1000 chip as the UWB hardware, which was manufactured by Decawave.

## 5. Machine Learning Models for Identification of the LOS, NLOS, and MP Conditions

Three machine learning models (SVM, RF, and MLP) were evaluated in this paper. SVM was regarded as a baseline model in the evaluation since it was the most commonly and frequently used model for the UWB-based identification of NLOS conditions in the literature [9,10,17,21]. The configuration and setup for each classifier are discussed in the subsequent subsections.

The training and test times of each classifier reported in this section were based on a single concurrent CPU core without using any parallel computing devices such as the GPU. The evaluation was done on the same machine for all classifiers. The reported results for all classifiers in this section (SVM in Section 5.1, RF in Section 5.2, and MLP in Section 5.3) were based on 10 iterations of randomly splitting the measured training, validation, and test data. The training and test datasets used in this section were the random splitting of the data collected in Figure 3b. This explained the test datasets collected in Figure 3c, which were used only in the evaluation results presented in Section 7. The extracted features used for all classifiers in this section were based on the discussion and selection observed in Section 6. The reported training and test times per sample (mean value) for each classifier were estimated in two steps. First, we estimated the total amount of time it took for the whole dataset in the training and test phases using the corresponding training and test dataset. Then, the measured time was divided by the total number of samples to get the mean value per sample.

Generally, several parameters in the ML classifiers were tuned to achieve the optimized results. Moreover, each classifier had its specific hyper-parameters, which were not compatible with one another. Therefore, a direct comparison using exactly the same parameters for all classifiers was impossible. For the sake of simplicity and better representation of the results, the comparison is done by choosing the most important and influential parameters for each classifier in this section. This implied that the kernel type was chosen for SVM, and the number of decision trees in the forest was selected for RF. For MLP, the number of hidden layers including the total number of neurons in each layer was evaluated to choose the best option for the given problem.

For the reproducible results, the parameters for each classifier such as the activation function, optimizer, earlier stopping criteria for the training, learning rate, etc., were based on the default setting of the scikit-learn [26] library if nothing is explicitly mentioned in the following. The applied stable version of the scikit-learn library was 0.23.1 at the time of writing this paper.

### 5.1. Support Vector Machine Classifier for the UWB Localization System

SVM is a supervised machine learning technique suitable for solving both classification and regression problems [43,44]. It is strongly based on the framework of statistical learning theory [45]. SVM has also been recognized as one of the most frequently used classification techniques in the machine learning community in the past due to its robustness and superior performance without the need to tune several parameters compared to deep neural networks [9]. In short, SVM takes the data as an input and determines a hyper-plane that separates the data into predefined classes. The hyper-plane was established in the SVM algorithm by maximizing the margin between the separable classes as wide

as possible. Table 2 presents the comparison of four kernel types in SVM using the UWB measurement data and extracted features examined in Section 4.

**Table 2.** Comparison of the SVM configurations based on the kernel functions.

| Kernel Types | Mean Accuracy with std (%) | Mean Training Time per Sample (ms) | Mean Test Time per Sample (ms) |
|---|---|---|---|
| **Radial basis function (RBF)** | **82.96** $\pm$ 0.14 | 2.06 $\pm$ 0.18 | 0.99 $\pm$ 0.01 |
| Linear function | 72.59 $\pm$ 0.25 | **1.92** $\pm$ 0.08 | **0.53** $\pm$ 0.01 |
| 3rd order polynomial function | 70.82 $\pm$ 0.19 | 3.05 $\pm$ 0.09 | 0.80 $\pm$ 0.02 |
| Sigmoid function | 50.59 $\pm$ 3.05 | 3.01 $\pm$ 0.27 | 1.59 $\pm$ 0.09 |

The bold text and numbers in the table refer to the chosen kernel type for the evaluation and the best performance scores for each metric respectively.

The choice of the kernel types in SVM had a strong influence on its accuracy regarding our particular measurement of UWB data. The results in Table 2 show that the radial basis function (RBF) kernel reached the highest accuracy with 82.96%, while the sigmoid function provided the poorest with 50.59%. Both linear and third-order polynomial functions had comparable results. In terms of training and test times, the linear function achieved the lowest time per sample while the sigmoid function showed the worst performance with the highest time per sample. In all circumstances, the training and test times in SVM were in the order of milliseconds. This meant that SVM had the poorest performance in terms of test time compared to RF (Section 5.2) and MLP (Section 5.3).

*5.2. Random Forrest Classifier for the UWB Localization System*

According to the original paper in [46], random forests (RF) are a combination of decision tree predictors in the forest such that each tree depends on the values of a random vector, which is sampled with the independent and identical distribution for all the trees. In brief, RF is built upon multiple decision trees and merges them to get a more accurate and stable prediction as its final output. Two significant advantages of RF are (i) the reduction in over-fitting by averaging several trees and (ii) the low risk of prediction error since RF typically makes the wrong prediction only when more than half of the base classifiers (decision trees) are wrong. The disadvantage, though, is that RF is typically more complex and computationally expensive than the simple decision tree algorithm. In general, the more trees in the forest, the better the prediction. However, this flexibility comes with the cost of the processing time (training and test times), as described in Table 3.

**Table 3.** Comparison of the RF configurations based on the numbers of decision trees in the forest.

| No. of Decision Trees in the Forest | Mean Accuracy with std (%) | Mean Training Time per Sample (µs) | Mean Test Time per Sample (µs) |
|---|---|---|---|
| 5 decision trees | 90.91 $\pm$ 0.18 | **4.84** $\pm$ 0.25 | **1.26** $\pm$ 0.02 |
| 10 decision trees | 91.55 $\pm$ 0.09 | 9.38 $\pm$ 0.27 | 2.33 $\pm$ 0.08 |
| 20 decision trees | 91.83 $\pm$ 0.10 | 18.68 $\pm$ 0.52 | 4.66 $\pm$ 0.08 |
| 30 decision trees | 91.89 $\pm$ 0.11 | 27.30 $\pm$ 0.41 | 6.82 $\pm$ 0.13 |
| **50 decision trees** | 91.99 $\pm$ 0.11 | 45.44 $\pm$ 0.62 | 11.30 $\pm$ 0.27 |
| 100 decision trees | 92.07 $\pm$ 0.09 | 90.42 $\pm$ 1.39 | 22.43 $\pm$ 0.34 |
| 200 decision trees | 92.12 $\pm$ 0.13 | 179.85 $\pm$ 3.10 | 44.85 $\pm$ 0.47 |
| 500 decision trees | **92.13** $\pm$ 0.12 | 460.04 $\pm$ 10.27 | 113.39 $\pm$ 1.44 |

The bold text and numbers in the table refer to the chosen decision trees in the forest for the evaluation and the best performance scores for each metric respectively.

The prediction accuracy in RF increased steadily as the number of decision trees in the forest increased (Table 3). However, the improvement slowed down when the number of trees in the forest was more than 50 in this particular UWB data. In contrast, the training and test time keep increasing linearly by the increase of the decision trees in the forest. This implied that the training and test

times (the smaller the magnitude of the metric, the better the performance) were negatively affected by the growth of trees in the forest. Therefore, a trade-off between the accuracy by growing trees in the forest and the efficiency of the test time should be thoroughly made. In terms of training time, RF performed the fastest among the three classifiers compared to SVM (Section 5.1) and MLP (Section 5.3), i.e., the training time per sample in RF was in the order of microseconds.

### 5.3. Multi-Layer Perceptron Classifier for the UWB Localization System

MLP is a type of deep feedforward artificial neural networks, which contains at least three layers (an input layer, a hidden layer, and an output layer) in a single network [47]. Typically, the neurons in the hidden and output layers of the MLP use nonlinear activation functions such as sigmoid, ReLU, and Softmax. The term deep is usually applied when there is more than one hidden layer in the network. MLP utilizes the backpropagation algorithm [48] for training the network. In this paper, the MLP classifier is configured using the rectified linear unit (ReLU) as the activation function for the hidden layers, the Softmax function as the output layer, and the Adam (adaptive moment estimation) as an optimization algorithm. The maximum number of epochs was set to 500, allowing early stopping if the training loss had not improved for 10 consecutive epochs.

The evaluations of MLP were conducted for up to 6 fully-connected hidden layers in two conditions using 50 and 100 neurons in each hidden layer (Table 4). The results showed that there was a significant increase in the overall accuracy by adding a second and third hidden layer to the network. However, the improvement was meager when more than 3 hidden layers were used for a network. In terms of the number of neurons per layer in the network, the use of 100 neurons in each hidden layer constantly beat the use of 50 up to four consequent layers. However, the difference could not be clearly distinguished when more than 4 layers were used in the network.

In terms of the processing time (training and test times), adding more hidden layers and more neurons in the network had a negative impact on the performance (the last two columns in Table 4), i.e., the lower the processing time, the better the performance. Therefore, a trade-off between the accuracy and processing time was necessary to have an efficient performance. The results in Table 4 suggest that the use of 3 hidden layers in which each contained 100 neurons seemed a good choice for solving the evaluated UWB-based multi-class classification problem.

**Table 4.** Comparison of the MLP configurations based on the number of hidden layers and neurons.

| No. of Neurons in each Hidden Layers | No. of Hidden Layers | Mean Accuracy with std (%) | Mean Training Time per Sample (ms) | Mean Test Time per Sample (µs) |
|---|---|---|---|---|
| 50 | 1 | $84.93 \pm 0.26$ | **0.68** $\pm 0.27$ | **1.26** $\pm 0.06$ |
| | 2 | $88.77 \pm 0.53$ | $1.45 \pm 0.07$ | $3.23 \pm 0.55$ |
| | 3 | $90.45 \pm 0.51$ | $2.57 \pm 0.25$ | $5.46 \pm 0.20$ |
| | 4 | $90.95 \pm 0.35$ | $2.84 \pm 0.63$ | $8.79 \pm 0.41$ |
| | 5 | **91.04** $\pm 0.66$ | $3.50 \pm 1.07$ | $12.20 \pm 0.26$ |
| | 6 | $90.61 \pm 1.26$ | $3.70 \pm 0.74$ | $15.25 \pm 3.28$ |
| **100** | 1 | $85.88 \pm 0.12$ | **1.02** $\pm 0.01$ | **2.51** $\pm 0.05$ |
| | 2 | $89.78 \pm 0.59$ | $3.75 \pm 0.19$ | $12.20 \pm 1.53$ |
| | **3** | $91.36 \pm 0.54$ | $5.68 \pm 1.23$ | $18.12 \pm 1.71$ |
| | 4 | **91.38** $\pm 0.44$ | $5.08 \pm 1.67$ | $23.32 \pm 3.75$ |
| | 5 | $90.85 \pm 0.67$ | $7.90 \pm 2.60$ | $29.24 \pm 0.94$ |
| | 6 | $91.33 \pm 0.44$ | $9.42 \pm 3.90$ | $32.47 \pm 1.81$ |

The bold numbers in the table refer to the chosen configuration of MLP for the evaluation and the best performance scores for each metric.

### 5.4. Section Summary

In summary, RF was the fastest among the three methods for the given dataset in terms of the training time (Figure 4). This was because the training time per sample data in RF only took in the

order of microseconds. Meanwhile, SVM and MLP were in the order of several milliseconds depending on the configuration and setup. In terms of test time, SVM performed worst among the three with a test time in the order of milliseconds, while RF and MLP are in the order of microseconds.

Taking into consideration the evaluated results presented in this section, we established the configurations of three classifiers for further processing in Section 7. The summarized overview is represented in Figure 4. The selected configurations were the radial basis function kernel approach for SVM, 50 independent decision tree estimators for RF, and three hidden fully-connected layers with 100 neurons in each layer for the MLP network.

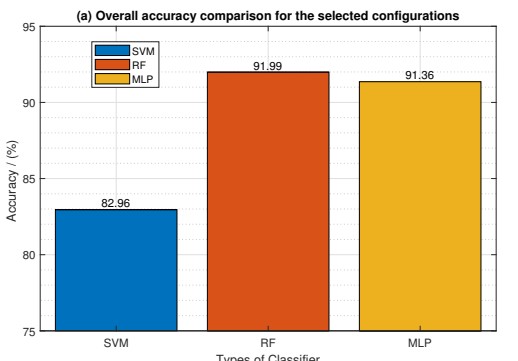 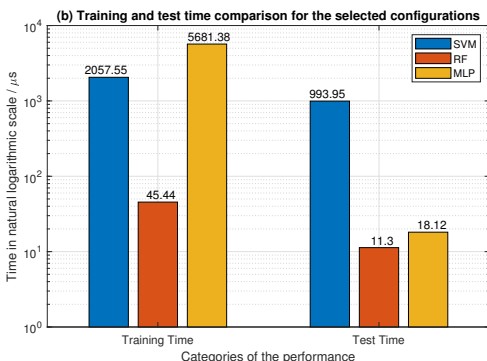

**Figure 4.** Summary of the chosen configurations for the three classifiers, which is the RBF kernel for SVM, 50 estimators for RF, and three hidden layers with 100 neurons in each for MLP. (**a**) Overall accuracy comparison for the selected configurations (**b**) Training and test time comparison for the selected configurations.

## 6. Data Preprocessing and Feature Selection

This section examines the impact of feature selections (Section 6.1) and features scaling, i.e., the standardization technique in this manuscript (Section 6.2), for the three evaluated ML models. The experimental results presented in Section 7 were performed based on the outcomes of this section.

### 6.1. The Impact of Feature Extraction in the Evaluated Machine Learning Models

Based on the extracted features defined in Section 4.3, the performance comparison of feature extractions for five categories is illustrated in Figure 5. The five categories were built upon: (i) 12 extracted features (i.e., the full features in our evaluation), (ii) 10 features excluding standard and maximum noises, (iii) 5 features, i.e., the reported distance, the CIR, and the first, second, and third harmonics of the FP, (iv) 3 features, i.e., reported distance, the CIR, and the first harmonics of FP, and (v) 2 features, i.e., the CIR and first harmonic of FP.

Regarding feature extractions of the UWB measurement data, Figure 5 indicates that a notable degradation in accuracy occurred for the three ML models when two features were applied in the evaluation. The rest of the categories (starting from three to 12 features) provided more or less comparable results. Moreover, we noticed during the evaluation that using the reported distance as a feature played an important role in feature extractions for UWB data. Furthermore, this was also the metric that we were most interested in estimating the position in UWB. We also observed that the contribution of the amplitudes of the three harmonics (first, second, and third) in the FP signal implied comparable impacts. This implied that picking any one of them as a feature provided an equivalent performance in the case of feature reductions. The amplitude of the CIR is undoubtedly an important feature in UWB, which represents a vital role in the identification of NLOS in the conventional technique using (2).

The empirical results in Figure 5 suggested that the most suitable choice for the evaluation in terms of minimum features and optimal performance was to use three features. This was because there were no significant gaps between the use of three to 12 features in terms of accuracy (Figure 5).

The accuracy in this context refers to the overall probability of the defined three classes, which was accurately estimated during the measurement, i.e., independent of the specific LOS, NLOS, and MP conditions. As a rule of thumb, fewer features in the model typically allow less computation and better resource efficiency, especially for MCU-based platforms. In terms of the test time, the performance of MLP degraded when less than five features were applied. However, there were no notable differences in SVM and RF when more than 2 features were used in the evaluation (Figure 5). Therefore, the experimental evaluation results presented in Section 7 were based on three features, specifically the reported distance, the amplitude of the CIR signal, and the amplitude of the first harmonics in the FP signal.

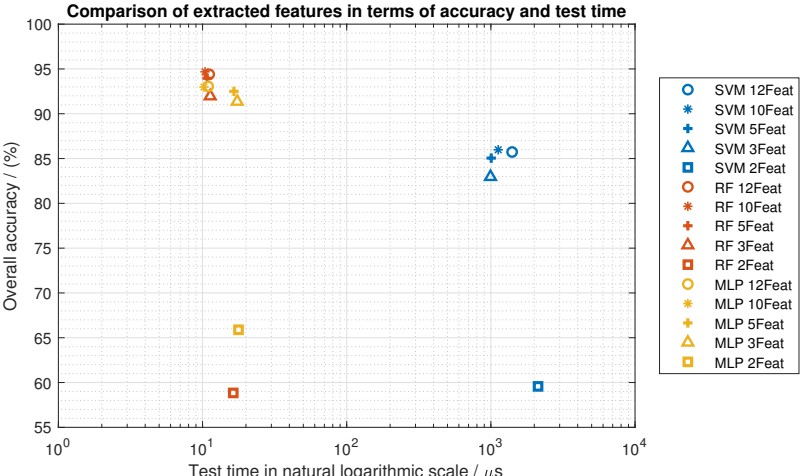

**Figure 5.** Performance comparison of three ML models (SVM, MLP, and RF) using different extracted features at training and test data collected for the same scenarios.

### 6.2. The Impact of Feature Scaling in the Evaluated Machine Learning Models

Feature scaling is a model dependent parameter in ML. It is a technique to normalize or standardize the range of independent variables or features of input measured data in a preprocessing step. This typically allows a faster training time and a better performance in many ML models. This section briefly reveals the effects of feature scaling in the three evaluated ML models. Besides, there exist ML models where their performance is not affected by the feature scaling in the preprocessing of the input data. A good representative of such a model in our evaluation was the RF classifier (Figure 6). In this paper, the feature scaling was performed using the standardization technique. This typically means rescaling the data in preprocessing to have a mean of zero and a standard deviation of one (unit variance).

Figure 6a depicts the impact of feature scaling for the three ML models in terms of the overall accuracy. Scaling the input data in the preprocessing phase had a notable impact on the SVM and MLP classifiers in terms of accuracy. In SVM, the overall accuracy was improved from 67.95% to 82.95% by scaling the features of the input data. Similarly, the accuracy of the MLP was increased from 85.70% to 91.36%. However, the RF gave equivalent outcomes in both scaled and unscaled features.

In terms of training time using three features in each model, SVM learned significantly faster when feature scaling was used (Figure 6b). To be precise, the training time of SVM reduced from $4195.14 \pm 210.38\,\mu\text{s}$ to $1977.12 \pm 185.45\,\mu\text{s}$ by scaling the features. However, RF and MLP did not show obvious improvement for training time in our small-scale three feature evaluation. Specifically, the training time of RF for the scaled and unscaled features was $44.97 \pm 2.38\,\mu\text{s}$ and $47.98 \pm 4.74\,\mu\text{s}$, respectively. Likewise, the training time of MLP for the scaled and unscaled features was $5562.66 \pm 1313.85\,\mu\text{s}$ and $5737.97 \pm 1949.05\,\mu\text{s}$, respectively.

In terms of test time, feature scaling hurt the performance of MLP (Figure 6b). Specifically, the value of the test time (the smaller, the better) in MLP degraded from $10.41 \pm 0.54\,\mu s$ to $17.58 \pm 1.27\,\mu s$ by feature scaling. On the contrary, the performance of the test time in SVM improved when feature scaling was applied, i.e., $1839.53 \pm 53.70\,\mu s$ for unscaled features and $956.28 \pm 13.94\,\mu s$ for scaled features. Again, RF did not show any significant improvements except a small variation in its standard deviation, which implied $11.48 \pm 0.99\,\mu s$ for unscaled features and $11.48 \pm 1.42\,\mu s$ for scaled features, respectively.

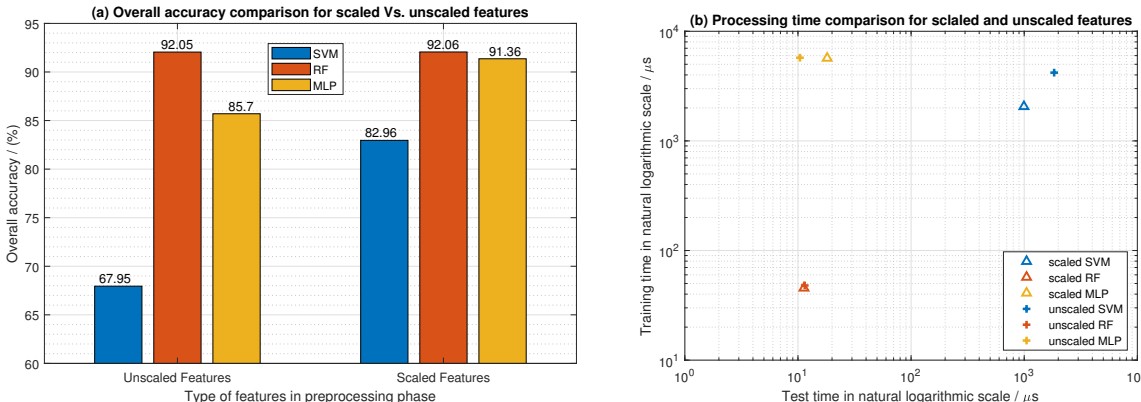

**Figure 6.** Comparison of the overall accuracy, training, and test times for the scaled vs. unscaled features in the preprocessing phase. (**a**) Overall accuracy comparison for scaled Vs. unscaled features (**b**) Processing time comparison for scaled and unscaled features.

## 7. Evaluation Results

This section examines the experimental evaluation results of three ML classifiers based on two quantitative metrics: (i) an F1-score, which was used to compare the performance of the three evaluated classifiers in this paper (Section 7.1), and (ii) a confusion matrix that gave an insightful representation of the reported results for each individual classifier (Section 7.2).

### 7.1. Performance Comparison of the Three Classifiers Using the Macro-Averaging F1-Score as a Metric

To give an overview of the actual state in each trial conducted 10 times, we use the macro-averaging F1-score to compare the performance of the three classifiers in this section. The F1-score, i.e., in contrast to the overall accuracy score in the confusion matrix (Section 7.2), is extensively used to quantify the classifier's performance in ML because it takes into account both the precision and recall to compute the decisive score [11,49]. It is the harmonic mean of the precision and recall, which can be expressed for a binary classification as:

$$F1 = 2 \cdot \frac{Precision \cdot Recall}{Precision + Recall} \tag{4}$$

For multi-class classification, there are two typical ways (macro-averaging and micro-averaging) to compute the overall F1-score for a classifier [49]. We applied the macro-averaging technique in our evaluation, which treated all the classes equally. Based on the mentioned macro-averaging F1-score, Figure 7 compares the experimental evaluation results of the three classifiers in two test environments (the scenario that was the same as vs. different from the training state) defined in Section 4.2.2. The solid lines denote the results of the test dataset when the data in the test state were collected in a different environment from the training state. The dotted lines show the results of the test dataset achieved from the same scenario as the training state.

In general, a significant gap between the two test scenarios was discovered in the experimental evaluation results of the three classifiers (Figure 7). The figure reveals that impressive outcomes

were achieved in RF and MLP classifiers when the test and training states were conducted in the same environments. However, the performance of SVM was relatively low in this scenario compared to RF and MLP. In contrast, the performance of all classifiers was notably degraded when the test environment was different from the training state. Specifically, the resultant mean of the SVM classifier based on the macro-averaging F1-score was reduced from 0.83 (when the training and test scenarios were the same) to 0.75 (when the training and test scenarios were different). Similarly, the performance of the RF decreased from 0.92 to 0.73. The MLP classifier showed a degradation from 0.91 to 0.72. The results showed that an immediate conclusion and judgment on the choice of classifiers based on a single test scenario or environment could be misleading. The core reason was that the estimated precision and characteristics of the measured wireless signal, i.e., the UWB ranging data in our evaluation, are affected by a variety of physical impacts in indoor environments, as previously mentioned several times in this manuscript.

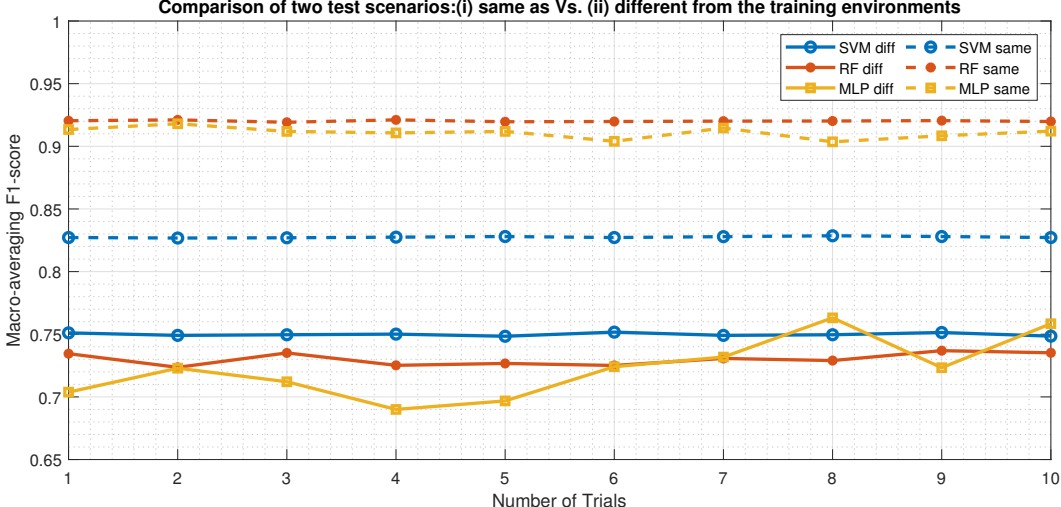

**Figure 7.** Performance comparison of the three evaluated classifiers based on the macro-averaging F1-score in two different scenarios: (i) dotted lines denote that the training and test data came from the same environment; (ii) solid lines denote that the training and test data came from different environments.

It is interesting to see that SVM stood out to be the best classifier in our evaluation when the test scenario was different from the training state (Figure 7). However, it had the poorest performance among the three classifiers when the same environments of training and test were applied. Moreover, the outcomes of SVM were consistent in all of the evaluated trials in both of the scenarios. Indeed, the outcomes of RF were also quite stable across the whole trials compared to the MLP classifier. However, many fluctuations were evident in the predicted outputs of the MLP, especially in the condition where the training and test environments were different.

In all experiments, the lowest F1-score was 0.69 in Trial No. 4 when MLP was used as a classifier, and the highest score reached 0.92 using RF. This outcome showed that the ML-based classifications, regardless of the type of the classifier, were more effective in the multi-class identification of UWB data than the traditional approaches described in Section 2.

### 7.2. Result Representation of the Three Evaluated Classifiers Using the Confusion Matrix

To examine the study more extensively, a comparative analysis of the two test scenarios for each classifier was conducted using the confusion matrix in this section. In the confusion matrix (Figures 8–10), the output class in the Y-axis refers to the prediction of the classifier, and the target class in the X-axis refers to the true reference class. The overall accuracy of the classifier is given in the bottom right corner of each confusion matrix. The last column in each category of

the confusion matrix indicates the precision (positive predictive value) and its counterpart the false discovery rate (FDR) of the classifier. Likewise, the last row in each category gives the recall (sensitivity or true positive rate) and its complement the false-negative rate (FNR). The correct predictions for each category are expressed in the diagonal of the confusion matrix. The values in the off-diagonal correspond to the Type-I and Type-II errors. For a scenario where training and test datasets were collected in different environments, the confusion matrices presented in this section were based on the mode of each classifier (i.e., we chose the most frequently predicted class in each trial as our estimator output). For a scenario where the training and test environments were the same, the confusion matrix was based on Trial No. 5 out of the 10 trials described in Section 7.1. The reason was that the random splitting of the test dataset for the true class in this scenario was different for each trial. Moreover, all trials in this scenario gave comparable results as reported in Figure 7.

### 7.2.1. Comparative Analysis of the Two Test Scenarios for SVM Classifier

An insightful comparison of the two test scenarios based on the confusion matrix for SVM is presented in Figure 8. The result showed that the overall accuracy of the SVM significantly dropped, i.e., from 82.8% to 75.4%, when the tested dataset was different from the training state. We observed that this was the cause of a significant decrease in the identification process of the MP condition. By comparing the two test scenarios in Figure 8a,b, the predicted accuracy of the MP conditions in SVM declined from 28.6% to 19.0%. Meanwhile, there existed no sharp deviations in the predicted accuracy of the LOS and NLOS conditions in both test scenarios. This increased the misclassification rate of the LOS and NLOS conditions as an MP in the SVM classifier.

To be precise, a significant misclassification rate of the MP condition as an NLOS was detected in the evaluated results, i.e., the value rose from 0.6% (when the training and test were in the same condition) to 5.8% (when the training and test were in different conditions), as provided in Figure 8. The misclassification rate of LOS as MP was also quite high, i.e., it increased from 4.2% to 8.5%. The main reason could be the data collected for MP conditions when the two transceivers were too close to each other. In that case, it was acceptable to interpret the received signal in MP condition as an LOS.

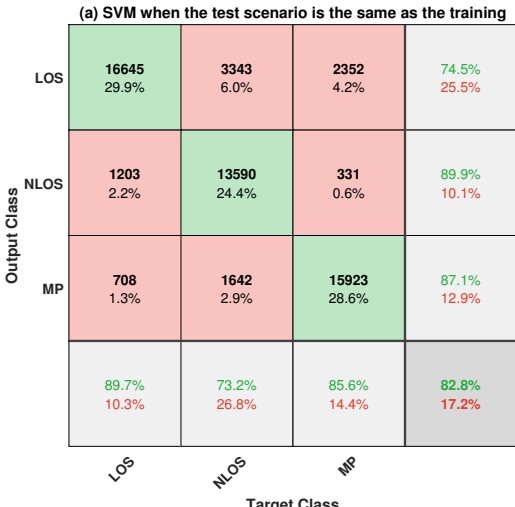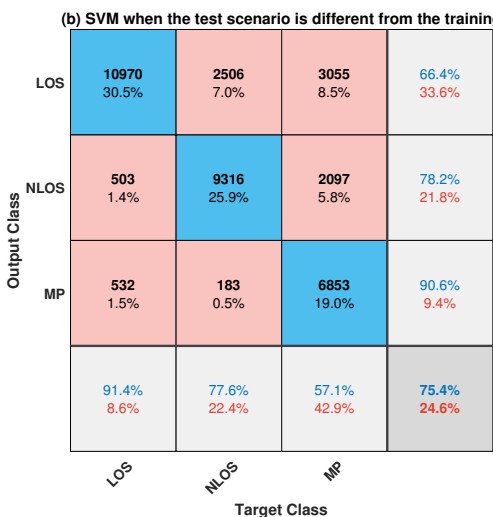

**Figure 8.** Comparison of the multi-label classification results for SVM using the confusion matrix in two different scenarios: (**a**) the test dataset was obtained from the same environments as in the training state; (**b**) the test dataset was collected in the environments different from the training state. In the evaluation, the radial basis function was used as the kernel for the SVM classifier.

Regarding the NLOS condition in both test scenarios, we observed quite a good outcome in the predicted accuracy, precision, and recall of the SVM classifier (Figure 8). This result was

crucial because the main impact on the performance of the UWB localization algorithm was the NLOS condition (Section 2). The misclassification of the LOS as an NLOS did not produce a severe consequence on the overall performance of the UWB system. The reason was that the location algorithm assigned different weights to the classification results, and giving LOS as a smaller weight did not hurt the system performance.

### 7.2.2. Comparative Analysis of the Two Test Scenarios for the RF Classifier

Similar to the SVM classifier, the overall accuracy of the RF classifier degraded strikingly from 91.9 to 73.5 when the test and training environments were different (Figure 9). Again, the cause of this significant decrease in RF was evident in the predicted accuracy of the MP conditions, i.e., it reduced from 30.4% to 19.6%. Unlike the SVM classifier, the predicted accuracy of the RF classifier in both the LOS and NLOS conditions noticeably declined as well. This implied the predicted accuracy of LOS degraded from 30.9% to 26.9% while NLOS decreased from 30.6% to 27.0%. This pushed the overall accuracy of the RF classifier to be worse than the SVM in the scenario where the training and test environments were different.

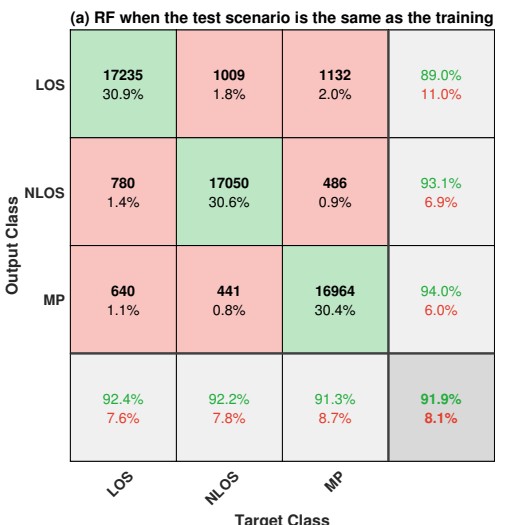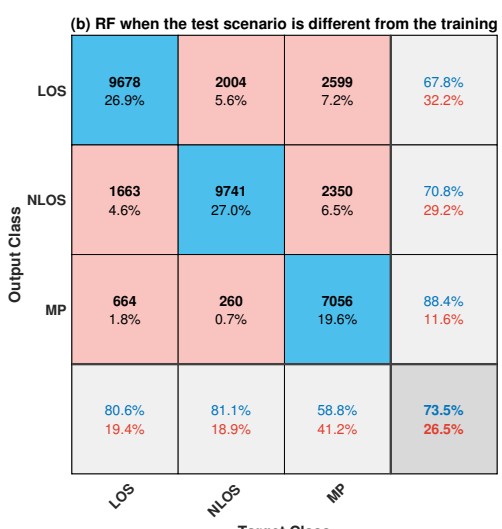

**Figure 9.** Comparison of the multi-class classification results for the RF classifier using the confusion matrices in two scenarios: (**a**) the test dataset obtained from the same conditions as the training data; (**b**) the test dataset collected in a different condition from the training. In the evaluation, fifty decision trees were used in the forest of the applied RF classifier.

Similar to the aforementioned reason and condition in the SVM classifier, the misclassification rate of the MP condition as either the LOS or NLOS was quite high in the RF classifier as well (Figure 9). To give the exact values, the misclassification rate of the MP condition as NLOS increased from 0.9% to 6.5%. Similarly, the misclassification of the MP as LOS grew from 2.0% to 7.2%. In the RF classifier, the misclassification rate of the NLOS as an LOS was also noticeably high, i.e., it increased from 1.8% to 5.6%. Again, the predicted accuracy of the NLOS condition in both of the two test scenarios was quite satisfying; that is, 30.6% out of 100% for the three classes in the scenario when the training and test data were in the same environments. Similarly, it was 27.0% when the data for the training and test environments were different.

### 7.2.3. Comparative Analysis of the Two Test Scenarios for the MLP Classifier

Figure 10 compares the multi-class classification results of two test scenarios using MLP as a classifier. Similar to the previously mentioned two classifiers, the overall accuracy of the MLP considerably declined from 91.2% to 72.9% when the test environments of the UWB were different from the training state. Repeatedly, this was caused by the false discovery rate of the MP conditions in

the measured UWB data. Specifically, the predicted accuracy of MP condition in the MLP classifier decreased from 30% to 19.8% out of 100% for the three classes. This outcome showed that the MP condition was the challenging class to identify throughout our evaluation in all classifiers (Figures 8–10).

The decrease of performance in a particular class causes an increase in the false discovery rate of other classes. Specifically, for our evaluation in MLP, the misclassification rate of MP as LOS increased from 2.2% to 7.7%, while MP as NLOS increased from 1.1% to 5.9% (Figure 10). In fact, the predicted accuracy of both LOS and NLOS also declined in the MLP classifier, i.e., from 30.9% to 26.7% in the LOS condition and from 30.2% to 26.4%.

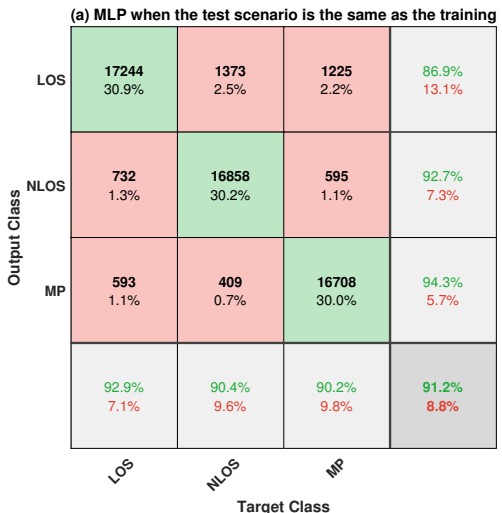 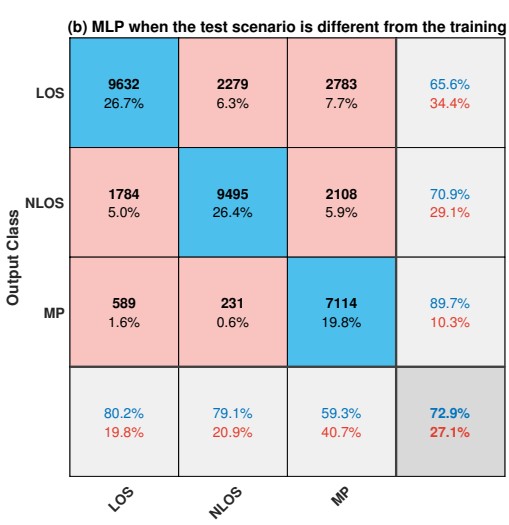

**Figure 10.** Comparison of the multi-class classification results for the MLP classifier using confusion matrices in two scenarios: (**a**) the test dataset obtained from the same conditions as the training phase; (**b**) the test dataset collected in a different condition from the training. In the evaluation, three fully-connected hidden layers with 100 neurons in each layer were used for the MLP classifier.

*7.3. Summary of the Experimental Evaluation Results*

In this paper, the multi-label classification results of the UWB data were quantified based on two metrics, i.e., the F1-score and confusion matrix. The F1-score is typically used in the literature to quantify the performance of ML-based classifiers because it provides a convenient single value score [11,42,49]. However, it can sometimes overlook the insightful information of some classes. Table 5 gives the typical summary of the two evaluated scenarios using both the F1-score and overall accuracy. The individual F1-score for each class (LOS, NLOS, and MP) is also given in the table.

**Table 5.** Summary of the results based on the F1-scores and overall accuracy.

| Scenarios | Classifiers | Individual F1-Scores | | | Macro-Averaging | Overall |
|---|---|---|---|---|---|---|
| | | LOS | NLOS | MP | F1-Scores | Accuracy (%) |
| Training and test environments are different | SVM | 0.77 | **0.78** | 0.70 | **0.75** | **75.35** |
| | RF | 0.74 | **0.76** | 0.71 | 0.73 | 73.52 |
| | MLP | 0.72 | **0.75** | 0.71 | 0.73 | 72.86 |
| Training and test environments are the same | SVM | 0.81 | 0.81 | **0.86** | 0.83 | 82.80 |
| | RF | 0.91 | **0.93** | **0.93** | **0.92** | **91.90** |
| | MLP | 0.90 | **0.92** | **0.92** | 0.91 | 91.20 |

The bold numbers in the table refer to the best performance scores for each class (LOS, NLOS, and MP) and each classifier based on the metrics and scenarios.

Based on the data given in Table 5, both the macro-averaging F1-score and overall accuracy showed that the SVM classifier gave the best performance when the training and test environments

were different. The consistent outcome in SVM also revealed the reason why it is one of the most frequently used classifiers in the literature [9,10,17,21]. In contrast, RF performed the best among the three evaluated classifiers regarding the same training and test environments.

However, it was hard to clarify using Table 5 that the predicted accuracy of MP condition was significantly low compared to the other two classes in all evaluated classifiers in a scenario when the training and test were in different environments. This phenomenon could be detected using the confusion matrix as previously described in Section 7.2.

## 8. Discussions

The identification process of the LOS, NLOS, and MP conditions in a wireless ranging system, especially in UWB, is crucial because they strongly influence the quality and accuracy of the actual measurement. For that reason, several contributions have been proposed in the literature to identify these conditions as already mentioned in Section 1. However, most of the contributions treated the issue as a binary classification problem (Section 2). To the best of our knowledge, only two papers [11,21] addressed the UWB-based classification process as a multi-class problem. In this paper, we defined three classes in UWB measurement data (LOS, NLOS, and MP conditions as presented in Section 2) and evaluated three ML classifiers (SVM, RF, and MLP as presented in Section 5) to identify these three defined classes.

Two metrics, F1-score and confusion matrix, were used for evaluating the performance of each classifier. Apart from these scores, the training and test times for each classifier were also given in our evaluation. As a matter of fact, this type of metric (training and test times) is typically ignored in the literature. However, it is undeniable that the magnitude of a test time in a certain classifier is usually vital in the overall performance of the system. Our results presented in Section 5 revealed that SVM had the poorest performance among the three classifiers in terms of the test time. In contrast, SVM gave the best performance in terms of the F1-score when different environments of training and test states were applied (Section 7). The two mentioned results showed us the interesting contradiction of two metrics in the SVM classifier, which could be an important factor in the system implementation of some ML applications.

The evaluation results based on the F1-score and overall accuracy pointed out that the measured environments in UWB had a strong effect on the performance of the three classifiers (Section 7). This referred to the striking degradation of the performance in all classifiers when different environments of training and test were applied in the evaluation. One may argue that this was caused by the overfitting of the model in the training state. However, this outcome occurred in all three evaluated classifiers regardless of the hyperparameters used in each classifier. Indeed, this was a part of the generalization problem caused by the inadequate representation of the conditions in the data. However, the data collection process in UWB, especially for the three mentioned classes, was quite time-consuming, elaborate, and costly because of the nature of the wireless signal. This meant the evaluated outcomes would be affected differently in several different ways by the conditions of the materials, types of walls, types of furniture, etc., in the measured environment. Our attempt was to provide the feasibility of the ML approach in the multi-class classification of the UWB measurement data in contrast to the conventional technique given in Section 2. The results based on the F1-score in our experiments showed 0.69 in the worst-case scenario and 0.92 in the best-case scenario. The outcomes were quite satisfying and promising.

In general, the classification results of UWB in the literature were usually reported using single value metrics, i.e., the overall accuracy score [21,23] and F1-score [11]. In some cases, the recall and true negative rate were applied in addition to the accuracy score, as in [38]. Typically, in a binary classification problem, the receiver operating characteristic (ROC) curve [50] and cumulative distribution function [9,18] are widely used. Those scores and metrics are particularly helpful for the comparative analysis of different classifiers. However, the insightful details of the actual conditions are usually overlooked or missed in some cases. Therefore, we used the complete confusion matrix in

this paper to examine the individual outcomes of each class in all evaluated classifiers (Section 7.2). Using the confusion matrix, we observed that the predicted accuracy of the MP condition significantly dropped in all classifiers when different environments of training and test were used in the data. This condition cannot be clearly evident using other above-mentioned metrics. This incident also proved that the identification of the MP condition was more challenging than the other two classes (NLOS and LOS). Moreover, we observed that the predicted accuracy of the NLOS condition was generally quite good in all of the evaluations regardless of the classifiers. At the same time, the misclassification rate of NLOS as either an LOS or MP condition was also moderately low. This criterion is crucial because the highest error rate in UWB measurements is usually caused by the NLOS condition (Section 2). In contrast, the misidentification of LOS as an NLOS condition in a certain model did not hurt the overall system performance of the UWB application.

## 9. Conclusions

In this paper, the multi-label (LOS, NLOS, and MP) identification for UWB ranging was conducted using three machine learning techniques (SVM, RF, MLP) as a classifier. This was in contrast to the typical binary classification approaches in the literature. The experimental evaluation results based on the F1-score proved that ML-based classifiers could identify the defined three classes with a high score, i.e., 0.69 in the worst-case scenario and 0.92 in the best-case scenario. However, it was unreasonable to single out the best classifier out of the three because their performances depended extensively on the environmental changes and the metrics used to quantify them. Moreover, the insightful results based on the confusion matrix revealed that the MP condition was the most challenging to identify among the three classes. It was also evident in the confusion matrices that the predicted accuracy of the NLOS condition was quite high throughout all the evaluated experiments.

As future work, the NLOS condition could be further classified into two classes based on the study conducted in [11]. This would lead the defined classes of UWB data to become four, i.e., LOS, MP, soft-NLOS, and hard-NLOS. Indeed, this also means much more data would need to be collected to identify these four conditions in several environments using several different materials and objects. Besides, the experimental evaluation will be conducted in the actual hardware of the microcontroller-based platform instead of a PC.

**Author Contributions:** Conceptualization, C.L.S.; methodology, C.L.S. and J.D.H.; validation, B.S., J.D.H., M.A., M.H., and U.R.; software, C.L.S., B.S., and M.A.; writing, original draft preparation, C.L.S.; writing, review and editing, B.S., J.D.H., M.A., M.H., and U.R.; visualization, C.L.S., J.D.H., M.A., and M.H.; project administration, M.H. and U.R.; supervision, U.R.; funding acquisition, U.R. All authors read and agreed to the published version of the manuscript.

**Funding:** We acknowledge support for the article processing charge by the Deutsche Forschungsgemeinschaft and the Open Access Publication Fund of Bielefeld University.

**Acknowledgments:** This work was supported by the Cluster of Excellence Cognitive Interaction Technology "CITEC" (EXC 277) at Bielefeld University, funded by the German Research Foundation (DFG). Author Cung Lian Sang was partly supported in this work by the German Academic Exchange Service (DAAD). The authors are responsible for the contents of this publication. We also sincerely thank the anonymous reviewers for their valuable suggestions and feedback.

**Conflicts of Interest:** The authors declare no conflict of interest. The funders had no role in the design of the study; in the collection, analyses, or interpretation of the data; in the writing of the manuscript; nor in the decision to publish the results.

## Abbreviations

The following abbreviations are used in this manuscript:

| | |
|---|---|
| AltDS-TWR | Alternative double-sided two-way ranging |
| BDT | Boosted decision tree |
| CIR | Channel impulse response |
| CNN | Convolutional neural network |
| FP | First-path |
| GP | Gaussian process |
| HSI | High speed internal (clock) |
| IoT | Internet of Things |
| KNN | K-nearest neighbor |
| LOS | Line-of-sight |
| MCU | Microcontroller unit |
| ML | Machine learning |
| MLP | Multi-layer perceptron |
| MP | Multi-path |
| NLOS | Non-line-of-sight |
| PATD | Preamble accumulation time delay |
| PUB | Publications at Bielefeld University |
| RBF | Radial basis function |
| RF | Random forest |
| RSS | Received signal strength |
| RX | Received or receiver |
| SNR | Signal-to-noise ratio |
| SVM | Support vector machine |
| TOF | Time-of-flight |
| UWB | Ultra-wideband |

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
