# Peer review of "Identification of NLOS and Multi-Path Conditions in UWB Localization Using Machine Learning Methods"

_applsci, doi:10.3390/app10113980_

Round 1
Reviewer 1 Report
The manuscript presents a classification based identification of LoS, NLoS and MP conditions for UWB Signals. The novelty of the paper is in the consideration of MP conditions because prior researches were focused on LoS and NLos cases. The used ML learning techniques are well known and they are applied properly. The goal and contribution of this research is clear. The used methodologies are sound. Therefore the acceptance of the manuscript for publication is suggested.
Author Response
Comments and Suggestions for Authors
The manuscript presents a classification based identification of LoS, NLoS and MP onditions for UWB Signals. The novelty of the paper is in the consideration of MP conditions because prior researches were focused on LoS and NLos cases. The used ML learning techniques are well known and they are applied properly. The goal and contribution of this research is clear. The used methodologies are sound. Therefore the acceptance of the manuscript for publication is suggested.
We would like to express our appreciation for your positive feedback.
Thank you very much for your encouraging comments and your valuable time for reviewing this manuscript.

Reviewer 2 Report
The paper presents a study on the use of classical machine learning algorithms to try to identify the type of UWB measurements according to their source. The authors define these possible sources as: LOS, Multipath, and NLOS. To complete this task, the authors carried out several measurement campaigns with commercial devices in real indoor environments. The idea and its development seem correct, although I have some doubts that I would need to see resolved before giving my total agreement.
1. The introduction defines the concepts LOS, NLOS, and MP that will be used later throughout the text. Although the concepts of LOS and NLOS have a clear meaning in this context, the MP class leaves me with some doubts. Especially when looking at figure 1. In this figure two "pyramids" are presented, "Anc1" and "Anc4", in this multipath situation. But these situations are clearly different. In "Anc1" the figure shows an obstacle that, I understand, totally blocks the direct line of sight and it causes the receiver to decode a rebound of the signal. Clearly, in this case, the error in the range will be high. However, for the "Anc4" case it seems that the multipath situation can also occur (according to this model) even if there is no obstacle between transmitter and receiver. Although multipath will obviously always appear, it is debatable whether its impact working with UWB can be even close to that of the "Anc1" case. After reading the rest of the paper, I understand that the "version" that has been analyzed in the paper is the one that corresponds to "Anc4" of figure 1. If this is the case, I would recommend redoing figure 1 to avoid confusion or increase the detail in the explanation.
2. In figure 2, the different scale makes it difficult to compare cases "a" and "b", with the results of "c". I would recommend unifying the scaling.
3. Figure 2d is the only figure that shows range error measurements of the whole article. How was the Ground truth calculated? How were these values calculated if the measurements were carried out randomly with the tag being moved by a person? On the other hand, the graph indicates that the "distance deviation" is negative for MP and NLOS, when the bias should always be positive in case of not being LOS above certain distances. Perhaps the meaning of this "distance deviation" should be better explained, if it is something different from what is intuitively understood (that is, the difference between the actual value and the value measured by the chip).
4. On the other hand, in view of the results of this figure 2d, the differences between the so-called LOS case and the MP case seem to be minimal. It shows a very similar, confidence interval? variance? (the legend should be expanded with more information about each mark in the graph) in both cases, and a very similar error average between both (barely 4cm difference). Such low differences may not be enough to consider them as two separate classes. If in the MP case the decoded signal had been a rebound instead of the original signal that would mean that A) the FP was totally blocked/attenuated by an obstacle and B) the signal rebounded off something very close to cause such a small error. This doesn't seem to be the case of the measuring scenarios and perhaps the differences observed have more to do with an attenuation of the signal energy that causes the detection algorithm of the DW1000 to introduce a different bias in the LOS and MP cases. One of Decawave's Application Notes might be of interest in this case: APS011 APPLICATION NOTE SOURCES OF ERROR IN DW1000 BASED TWO-WAY RANGING (TWR) SCHEMES. Perhaps the authors could discuss this possibility.
5. In line 168 a reference is cited that places the error for the MP case between a few centimeters and 60cm. The phrase "This can also be verified in our experimental evaluation presented in Figure 2 (d)" does not seem to be completely true, since in that graph the average error is less than 5 cm. Perhaps this sentence should be reformulated.
6. In section 4.2 it is stated that the data collected in the narrow corridors were intended to obtain measurements for the MP class, since the LOS cannot be distinguished by the multiple bounces of the signal. This statement seems a bit imprecise, especially considering the time resolution we are working with at UWB, even in the 3993MHz band. In a corridor like the one shown in figure 3b, at short distances it would be very difficult not to detect the original signal, while at long distances the delay differences between the original signal and a bounce would be very small. Again, the difference between the LOS and MP classes does not seem clear. On the other hand, in figure 3b it can be seen some sort of plank between the tags. Which scenario does this setup correspond to? Were the measurements obtained with this setup mixed with those obtained on foot? Is there some kind of ground truth that allows checking the error of each of the measurements with respect to the actual distance? In general, a more detailed description of the labeling process is missing.
7. Section 4.2.1 talks about a manual tagging of the measurements in blocks, but it gives little details about the differences between those blocks. How were the tags in each one of them? What kind of obstacles? Was the mobile tag always with the same orientation or was it rotated to some extent? If so, was there a limit to the rotation allowed?
8. In Tables 2, 3, and 4, which training and test sets were used to obtain these accuracy values? The whole measurements of the different sets? This is important to note, as the same hyperparameters are used to set up the different algorithms for all the different scenarios. In addition, it is also important to be able to analyze the results obtained, since for example RF is an algorithm that can tend to overfitting, which is perhaps what is happening in view of the results shown in Figure 4.
9. In line 459 it is explained that the reason for the drop in performance when applying the classifiers with a set of measurements captured in another scenario is "the quality of the wireless signal". Perhaps this phrase should be rephrased as it seems to imply that there are "better" and "worse" measurements, when they simply have substantially different values for the chosen features.
10. In line 462 when talking about SVM as the best algorithm for the case where training and test sets are captured in different scenarios, could not it indicate some overfitting in the case of the other algorithms when testing and training take place with samples from the same scenario? Perhaps some reflection could be added in this regard.
Author Response
Comments and Suggestions for Authors
The paper presents a study on the use of classical machine learning algorithms to try to identify the type of UWB measurements according to their source. The authors define these possible sources as: LOS, Multipath, and NLOS. To complete this task, the authors carried out several measurement campaigns with commercial devices in real indoor environments. The idea and its development seem correct, although I have some doubts that I would need to see resolved before giving my total agreement.
We would like to gratefully thank you for your valuable comments, feedback, efforts, and time for reviewing this manuscript. We revised the manuscript by addressing your valuable comments as follows.
- The introduction defines the concepts LOS, NLOS, and MP that will be used later throughout the text. Although the concepts of LOS and NLOS have a clear meaning in this context, the MP class leaves me with some doubts. Especially when looking at figure 1. In this figure two "pyramids" are presented, "Anc1" and "Anc4", in this multipath situation. But these situations are clearly different. In "Anc1" the figure shows an obstacle that, I understand, totally blocks the direct line of sight and it causes the receiver to decode a rebound of the signal. Clearly, in this case, the error in the range will be high. However, for the "Anc4" case it seems that the multipath situation can also occur (according to this model) even if there is no obstacle between transmitter and receiver. Although multipath will obviously always appear, it is debatable whether its impact working with UWB can be even close to that of the "Anc1" case.
We would like to thank you for your detailed comments regarding the illustration of Figure 1 in the manuscript. As we mentioned in the introduction section of line 38, the figure is intended to give the reader an abstract view of the presented three classes from a broader perspective. Because of the nature of the wireless signal, it will always be debatable to exactly extract either a certain MP condition is caused by bounded signals or non-obstructed multiple reflections such as walls by looking at the signal statistics. In either case, MP conditions cause errors in the measurement compared to LOS. Therefore, we assume that it is acceptable to define both the above-mentioned states as an MP condition if that situation could be identified using some methods as we try to do it so in this manuscript. To avoid the confusion, we added the following more explanation about MP condition in the revised manuscript as your suggestion.
“In Figure 1, we defined two possible MP conditions in wireless communication. The first condition is clear because the first path signal is completely blocked by the obstacle and the only received signal in the measurement is based on the bounded signal from the transmitter. However, distance measurement in wireless communication could also be distorted by multiple reflections of signals even if there is no direct obstacle between the transceivers. For instance, wireless measurement is conducted in a narrow corridor, tunnel, etc. We have confirmed the error caused by such MP conditions using UWB in our previous work [8]. Similar results based on UWB were also reported in [12].”
After reading the rest of the paper, I understand that the "version" that has been analyzed in the paper is the one that corresponds to "Anc4" of figure 1. If this is the case, I would recommend redoing figure 1 to avoid confusion or increase the detail in the explanation.
It is exactly true that we analyzed the MP condition corresponding to the multiple reflections of walls in this manuscript. The main reason is that MP condition can easily be detected and defined without the need to have a specific controlled environment for a wireless measurement campaign. In contrast, it is necessary to have a fully controlled environment in order to clearly separate the differentiation between the bounded and unobstructed MP conditions in real-world. Even in that scenario, the material used in the mentioned controlled environment will still have large impacts on the measurement data. To ease this constraint, we simply used the corridor which we know for sure that MP conditions would occur in this manuscript. As already mentioned above, we added the following more explanation in our revised manuscript as your suggestion. We also redrew Figure 1 as your suggestion.
“In Figure 1, we defined two possible MP conditions in wireless communication. The first condition is clear because the first path signal is completely blocked by the obstacle and the only received signal in the measurement is based on the bounded signal from the transmitter. However, distance measurement in wireless communication could also be distorted by multiple reflections of signals even if there is no direct obstacle between the transceivers. For instance, wireless measurement is conducted in a narrow corridor, tunnel, etc. We have confirmed the error caused by such MP conditions using UWB in our previous work [8]. Similar results based on UWB were also reported in [12].”
- In figure 2, the different scale makes it difficult to compare cases "a" and "b", with the results of "c". I would recommend unifying the scaling.
Thank you very much for pointing out the scaling in Figure 2. We have unified the scaling of all the three plots as your suggestion.
- Figure 2d is the only figure that shows range error measurements of the whole article. How was the Ground truth calculated? How were these values calculated if the measurements were carried out randomly with the tag being moved by a person?
We would like to thank you for asking these questions. Figure 2(d) stands for the motivational purpose of conducting multiclass classification of UWB localization in this manuscript. The ground truth distances were measured in our evaluation using a laser distance meter, CEM iLDM-150 model, which has an accuracy of millimeter ranges according to the manufacturer. Moreover, the results presented in Figure 2(d) were measured at fixed static scenarios (i.e. they are not in a dynamic/ moving scenario) as we mentioned in the caption of Figure 2(d). As you clearly point out, it is impossible to get the ground truth value of a random movement without using a highly accurate controlled environment. To avoid the confusion, we added the further explanation in the revised version of the manuscript as follow:
“It should be noted that the measured distances in Figure 2 (d) were conducted in a static scenario at approximately 6 m distances between Anchor and Tag for the three classes. The ground truth references of the distance were measured using a laser distance meter, CEM97iLDM-150 model (http://www.cem-instruments.in/product.php?pname=iLDM-150), which provides an accuracy of ±1.5 mm according to the datasheet of the manufacturer.”
On the other hand, the graph indicates that the "distance deviation" is negative for MP and NLOS, when the bias should always be positive in case of not being LOS above certain distances. Perhaps the meaning of this "distance deviation" should be better explained, if it is something different from what is intuitively understood (that is, the difference between the actual value and the value measured by the chip).
Thank you very much for pointing out the indication of “distance deviation” in Figure 2(d). This is exactly the distance deviation commonly understood in literature as you correctly mentioned. In this case, we subtracted the value measured by the chip from the value achieved by the laser distance meter, i.e. our reference actual value. In fact, the values measured by the chip are always greater than the ground truth as you precisely mentioned. To avoid the confusion, we reversed our subtraction, i.e. subtract the actual ground value from the measured value by the chip in Figure 2(d) of our revised version as your suggestion.
- On the other hand, in view of the results of this figure 2d, the differences between the so-called LOS case and the MP case seem to be minimal. It shows a very similar, confidence interval? variance? (the legend should be expanded with more information about each mark in the graph) in both cases, and a very similar error average between both (barely 4cm difference). Such low differences may not be enough to consider them as two separate classes.
We would like to thank you for pointing out the error difference between LOS and MP cases in Figure 2(d). The result of MP conditions in Figure 2(d) was based on a scenario when there is no obstacle between the two transceivers at 6m distances of ground truth. This means the distances between all of the three classes (LOS, NLOS, and MP) in the figure are at 6m distances in order to be the same condition for all cases. Because of this short distance between the two transceivers, the error differences between the LOS and MP conditions seem to be small. Moreover, we would like to mention that this error corresponds to a single range (a Tag to an Anchor) in the measurement. In general, at least three ranges (usually more in multilateration) of such measures are necessary for UWB localization. This means the combination of such errors in a ranging phase contributes to a significant impact on the overall system performance. To avoid the confusion, we added the following more explanation in our revised version of the manuscript:
“The result of MP conditions in Figure 2(d) was based on a scenario when there is no obstacle between the transceivers as illustrated in Figure 1 (section 1). To be precise, the measurement was conducted in indoor environments where multiple reflections from walls occurred in a narrow corridor.”
If in the MP case the decoded signal had been a rebound instead of the original signal that would mean that A) the FP was totally blocked/attenuated by an obstacle and B) the signal rebounded off something very close to cause such a small error.
We fully agree with the mentioned two statements regarding the MP conditions.
This doesn't seem to be the case of the measuring scenarios and perhaps the differences observed have more to do with an attenuation of the signal energy that causes the detection algorithm of the DW1000 to introduce a different bias in the LOS and MP cases. One of Decawave's Application Notes might be of interest in this case: APS011 APPLICATION NOTE SOURCES OF ERROR IN DW1000 BASED TWO-WAY RANGING (TWR) SCHEMES. Perhaps the authors could discuss this possibility.
Thank you very much for providing the application note from Decawave and for directing the likely biases caused by the DW1000 chip itself. Indeed, we did considered the biases mentioned in the application note and calibrated it accordingly in the evaluated transceivers before the measurement campaign. Moreover, that bias can be seen even in a purely LOS scenario as described in the application note itself. We also have a more rigorous evaluation of this matter in our previous work, which is reference [8] in the manuscript. Our observation reveals that the error occurred in MP case is different from the bias mentioned in the provided application note. One explanation that we can give based on our previous work is because of the preamble accumulation time delay (PATD) in the coherent receiver of UWB chip. To quote it from our previous work in which related references are given, “PATD is influenced by the presence of a multipath and quick frame arrival time because of a relatively short distance measurement. It is more significant when the reflected signal arrives within the chip period of the first path signal.” We added more information about this matter in the revised manuscript in accordance with your suggestion as follows:
“The result of MP conditions in Figure 2(d) was based on a scenario when there is no obstacle between the transceivers as illustrated in Figure 1 (section 1). Indeed, the measurement was conducted in indoor environments where multiple reflections from walls occurred in a narrow corridor.”
- In line 168 a reference is cited that places the error for the MP case between a few centimeters and 60cm. The phrase "This can also be verified in our experimental evaluation presented in Figure 2 (d)" does not seem to be completely true, since in that graph the average error is less than 5 cm. Perhaps this sentence should be reformulated.
We would like to thank you for your suggestion to reformulate the sentence. We have reformulated the sentence as follows:
“Similar deviation of the error in MP condition can be seen in our experimental evaluation presented in Figure 2 (d).”
- In section 4.2 it is stated that the data collected in the narrow corridors were intended to obtain measurements for the MP class, since the LOS cannot be distinguished by the multiple bounces of the signal. This statement seems a bit imprecise, especially considering the time resolution we are working with at UWB, even in the 3993MHz band.
We would like to thank you for your detailed comments. We would like to express that the error in MP conditions using UWB at the narrow corridor was evaluated and gave the results in our previous work [8]. Similar results were also reported in [12].
In a corridor like the one shown in figure 3b, at short distances it would be very difficult not to detect the original signal, while at long distances the delay differences between the original signal and a bounce would be very small. Again, the difference between the LOS and MP classes does not seem clear.
Regarding your valuable comments about short distances, we evaluated range error estimation using UWB at a closed LOS scenario (0.25 – 2 meters) and MP scenario at narrow corridor (4 – 24 meters) in our previous work [8]. All the measurements in that works are based on static scenarios using ground truth reference measured by laser distance meter. The results showed that distance error in MP condition is significant even in 4 m. The closed LOS scenario has different error characteristics. The research data of that work are also publicly available for those who are interested in exploring further. For a clearer explanation, we added the following sentences in the revised manuscript as your suggestion:
“In Figure 1, we defined two possible MP conditions in wireless communication. The first condition is clear because the first path signal is completely blocked by the obstacle and the only received signal in the measurement is based on the bounded signal from the transmitter. However, distance measurement in wireless communication could also be distorted by multiple reflections of signals even if there is no direct obstacle between the transceivers. For instance, wireless measurement is conducted in a narrow corridor, tunnel, etc. We have confirmed the error caused by such MP conditions using UWB in our previous work [8]. Similar results based on UWB were also reported in [12].”
On the other hand, in figure 3b it can be seen some sort of plank between the tags. Which scenario does this setup correspond to?
This scenario corresponds to the NLOS condition using pieces of concrete block at static scenario. We added the following sentence in the revised manuscript to be clearer as you pointed out.
“Besides, a thick concrete wall, pieces of concrete block, and a mixture of wooden and metal were also applied as parts of the obstacles for NLOS conditions in the two small rooms and their environments.”
Were the measurements obtained with this setup mixed with those obtained on foot?
Yes, the presented measurements were the mixture of both the static scenario and dynamic scenario (on foot).
Is there some kind of ground truth that allows checking the error of each of the measurements with respect to the actual distance?
For a static scenario, the ground truth distances were measured using a laser distance meter, CEM iLDM-150 model, as we have already answered in your comments above. However, it is impossible to get the ground truth value for a random dynamic movement scenario without using a highly accurate controlled environment. Therefore, we are sorry to tell you that we don't have any actual distances for checking random dynamic movement in our evaluation. Our attempt in this paper is to classify the scenarios (LOS, NLOS, or MP condition) based on the signal statistics and not finding distance errors in the measurement. Therefore, we assume that providing the actual distances is not necessary even though they are great important indeed. Moreover, we'll provide publicly all the data presented in this paper for further exploration for those who interested in it.
In general, a more detailed description of the labeling process is missing.
We would like to thank you for your suggestion. We added the following sentences in the revised manuscript to give more explanation about the labelling process as your suggestion:
“The block refers to a collection of data belonging to the same class that is saved separately into the computer as a single file. For instance, we collected the random movement data in corridor 2 (Figure 3) for 5 min using a data update rate of 20 Hz. At the same time, we made sure that there were no obstacles between the two transceivers, and the antenna is held still at \ang{90} during the process. Then, we annotated the block of this whole measured data as an MP class to be used in our evaluation. The same procedure was applied for the LOS and NLOS conditions in all measurement scenarios.”
- Section 4.2.1 talks about a manual tagging of the measurements in blocks, but it gives little details about the differences between those blocks.
We would like to thank you for pointing out the manual annotation process in our data. During the measurement campaign, we collected experimental data as a block for each scenario and class. The block refers to a collection of data belonging to the same class that is saved separately into the computer as a single file. For instance, we collected the random movement data in corridor 2 in Figure 3 for 5 min using data update rate of 20 Hz. At the same time, we made sure that there were no obstacles between the two transceivers, and the antenna is held still at \ang{90} during the process. Then, we annotated the block of this whole measured data as an MP class to be used in our evaluation. Regarding this, we added the following sentences in our revised manuscripts:
“The block refers to a collection of data belonging to the same class that is saved separately into the computer as a single file. For instance, we collected the random movement data in corridor 2 (Figure 3) for 5 min using a data update rate of 20 Hz. At the same time, we made sure that there were no obstacles between the two transceivers, and the antenna is held still at \ang{90} during the process. Then, we annotated the block of this whole measured data as an MP class to be used in our evaluation. The same procedure was applied for the LOS and NLOS conditions in all measurement scenarios.”
How were the tags in each one of them?
We did mentioned the condition of devices in our submitted manuscript at line 218. “In all cases, the data were collected for both static and dynamic conditions. In the dynamic case, the device attached to the PC stays static while another device was held by a human at random walks.” To give more explanation, we added the following sentences in our revised manuscript:
“In the static scenario, the two transceivers were at a vertical position in \ang{90} pointing the antenna of the DWM1000 module as an upward position without any rotation. However, the device held by a human during the dynamic condition was allowed to rotate between \ang{0} to \ang{180} in some cases of the data collection process.”
What kind of obstacles?
We did mentioned the types of obstacles we used in our submitted manuscript at line 220. “Moreover, the NLOS conditions by blocking the communication between two transceivers using a human as an obstacle were conducted in all cases depicted in Figure 3.” In addition, a thick concrete wall, a piece of square concrete, and a mixture of wooden and metal were also applied as parts of the obstacles for NLOS conditions in the two small rooms and their environments. To avoid the confusion, we added the following sentences in the revised manuscript:
“Besides, a thick concrete wall, pieces of concrete block, and a mixture of wooden and metal were also applied as parts of the obstacles for NLOS conditions in the two small rooms and their environments.”
Was the mobile tag always with the same orientation or was it rotated to some extent? If so, was there a limit to the rotation allowed?
Thank you very much for pointing this out. In static scenario, the two transceivers were at 90 degree without any rotation. However, the device help by a human during the dynamic condition is allowed to rotate in some of our experimental data between 0 to 180 degrees. To give more explanation about it, we added the following sentences in our revised manuscript as your suggestion:
“In the static scenario, the two transceivers were at a vertical position in \ang{90} pointing the antenna of the DWM1000 module as an upward position without any rotation. However, the antenna of the device held by a human during the dynamic condition was randomly rotated between \ang{0} to \ang{180} in some cases of the data collection process.”
- In Tables 2, 3, and 4, which training and test sets were used to obtain these accuracy values? The whole measurements of the different sets? This is important to note, as the same hyperparameters are used to set up the different algorithms for all the different scenarios.
Thank you very for asking this question. The training and test sets used in Tables 2, 3, and 4 were based on the data collected in Figure 3 (b). Regarding this, the test data sets collected in Figure 3 (c) were applied only in the evaluation results presented in section 7. In the former scenario (data based on Figure 3(b)), we randomly split the 30% of the data for the test sets. We repeated this manner for 10 times and the results achieved from these 10 iterations were presented in the mentioned three tables. We described this scenario in line 291. We added the following more information about this scenario in the revised manuscript as your suggestion:
“The training and test data sets used in this section were based on the random splitting of the data collected in Figure 3 (b). This explains the test data sets collected in Figure 3 (c) were used only in the evaluation results presented in section 7”.
In addition, it is also important to be able to analyze the results obtained, since for example RF is an algorithm that can tend to overfitting, which is perhaps what is happening in view of the results shown in Figure 4.
Thank you very much for your valuable comments. As we mentioned above, we randomly split the 30% of the data for the test sets in each iteration. We repeated this manner for 10 times and the results achieved from these 10 iterations were presented in the tables of this section and in Figure 4.
- In line 459 it is explained that the reason for the drop in performance when applying the classifiers with a set of measurements captured in another scenario is "the quality of the wireless signal". Perhaps this phrase should be rephrased as it seems to imply that there are "better" and "worse" measurements, when they simply have substantially different values for the chosen features.
Thank you very much for your suggestion to rephrase the usage in the manuscript. We rephrased the sentence according to your suggestion as follows:
“The core reason is that the estimated precision and characteristics of the measured wireless signal, i.e. the UWB ranging data in our evaluation, are affected by a variety of physical impacts in indoor environments, as previously mentioned several times in this manuscript.”
- In line 462 when talking about SVM as the best algorithm for the case where training and test sets are captured in different scenarios, could not it indicate some overfitting in the case of the other algorithms when testing and training take place with samples from the same scenario? Perhaps some reflection could be added in this regard.
We would like to thank you for mentioning the chances of overfitting in the evaluation when the training and testing take place in the same scenario. Regarding this, we actually gave the reflection of the matter in the discussion section of the manuscript in line 581 as in the following. Indeed, the discussion of this matter is not in the exact section where the esteemed reviewer has pointed out because we would like to give the issue from a broader point of view.
“One may argue that it is caused by the overfitting of the model in the training state. However, this outcome occurred in all of the three evaluated classifiers regardless of the hyper-parameters used in each classifier. Indeed, it is a part of the generalization problem caused by the inadequate representation of the conditions in the data. However, the data collection process in UWB, especially for the mentioned three classes, is quite time-consuming, elaborate, and costly because of the nature of the wireless signal. This means the evaluated outcomes will be affected differently in several different ways by the conditions of materials, types of walls, types of furniture, etc. in the measured environment. Our attempt is to provide the feasibility of the ML approach in the multi-class classification of the UWB measurement data in contrast to the conventional technique given in section 3.1.”

Reviewer 3 Report
In this work, the authors propose an experimental-based analysis of channel status conditions (i.e., LOS/NLOS/MP) in UWB communications. The key idea is to apply machine learning on relevant statistical features from a commercial chipset (DecaWave).
The work is very nice, I really appreciated it; it is well-motivated and writing is nice; the literature review is relevant. I have the following comments.
- In Fig. 1, I did not totally understand the case (d). Please elaborate a little bit.
- I'm wondering if the choice of human placement to emulate a (strong) NLOS condition is sufficient. Perhaps a wall or other material should be considered. Please at least comment on this.
- The considered features on page 9 can be easily extracted from the DW1000 chip. However, it would be nice if you can comment on their availability on other devices (even if Decawave is a sort of de-facto standard).
- Regarding the features, it would be nice if you can comment more on how they should be under different channel conditions. Something is stated in Fig. 1, but you can elaborate a little bit here.
- The rationale behind the various feature subsets in Fig. 5 should be better explained. How did you choose the features in the subset and why did you eliminate others?
- Please define the accuracy in Fig. 5-6. Is the probability that the channel condition is accurately estimated (independently of the specific LOS/NLOS/MP)?
- Can you please compare the ML-inspired approaches with others in the literature?
Author Response
Comments and Suggestions for Authors
In this work, the authors propose an experimental-based analysis of channel status conditions (i.e., LOS/NLOS/MP) in UWB communications. The key idea is to apply machine learning on relevant statistical features from a commercial chipset (DecaWave).
The work is very nice, I really appreciated it; it is well-motivated and writing is nice; the literature review is relevant. I have the following comments.
We would like to thank you for your encouraging comments, positive feedback, and your valuable time for reviewing this manuscript.
- In Fig. 1, I did not totally understand the case (d). Please elaborate a little bit.
We would like to thank you for pointing out the inadequate clarity regarding Figure 1 in the manuscript. We added the following more explanation about the figure in the revised version as your suggestion. Moreover, we modified Figure 1 for better clarity as your suggestion.
“In Figure 1, we defined two possible MP conditions in wireless communication. The first condition is clear because the first path signal is completely blocked by the obstacle and the only received signal in the measurement is based on the bounded signal from the transmitter. However, distance measurement in wireless communication could also be distorted by multiple reflections of signals even if there is no direct obstacle between the transceivers. We have confirmed the error caused by such MP conditions using UWB in our previous work [8]. Similar results were also reported in [12].”
- I'm wondering if the choice of human placement to emulate a (strong) NLOS condition is sufficient. Perhaps a wall or other material should be considered. Please at least comment on this.
We would like to thank you for pointing out the obstacle used in the evaluation. In fact, we also used a few other materials illustrated in Figure 3 in the evaluation when the measurement was conducted in the two small rooms and their environments. However, for the long corridors and hall, we did use only human as an obstacle for NLOS conditions. Please kindly excuse us for not clearly mentioning this scenario with texts in our submitted manuscript. To be clearer, we added the following sentences in the revised manuscript:
“Besides, a thick concrete wall, pieces of concrete block, and a mixture of wooden and metal were also applied as parts of the obstacles for NLOS conditions in the two small rooms and their environments.”
- The considered features on page 9 can be easily extracted from the DW1000 chip. However, it would be nice if you can comment on their availability on other devices (even if Decawave is a sort of de-facto standard).
We would like to thank you for enlightening us to give the perspective on other devices other than DW1000 chip. To be honest, we haven’t worked before with other UWB chips except for DW1000 in our previous experimental evaluation. However, we are quite certain that 9 out of the mentioned 12 features can easily be extracted from every UWB chips available in the markets. The remaining specific features related to DW1000 device that we mentioned in our manuscript are preamble accumulation count (feature no. 7), standard and maximum noise (feature no. 11 and 12). We added the following paragraph in the revised manuscript as your suggestion:
“Regarding the above mentioned 12 features, we would like to mention that the feature extraction is solely based on the DW1000 chip as a UWB hardware, which is manufactured by Decawave.”
- Regarding the features, it would be nice if you can comment more on how they should be under different channel conditions. Something is stated in Fig. 1, but you can elaborate a little bit here.
We would like to thank you for pointing out the different channel conditions in the manuscript. We have added the following sentences in the revised version as your suggestion:
“In Figure 1, we defined two possible MP conditions in wireless communication. The first condition is clear because the first path signal is completely blocked by the obstacle and the only received signal in the measurement is based on the bounded signal from the transmitter. However, distance measurement in wireless communication could also be distorted by multiple reflections of signals even if there is no direct obstacle between the transceivers. We have confirmed the error caused by such MP conditions using UWB in our previous work [8]. Similar results were also reported in [12].”
“The result of MP conditions in Figure 2(d) was based on a scenario when there is no obstacle between the transceivers as illustrated in Figure 1 (section 1). Indeed, the measurement was conducted in indoor environments where multiple reflections from walls occurred in a narrow corridor.”
- The rationale behind the various feature subsets in Fig. 5 should be better explained. How did you choose the features in the subset and why did you eliminate others?
We would like to thank you for pointing out the need of better explanation in Fig 5. We added the following sentences in the revised manuscript as your suggestion. Moreover, we also modified Figure 5 to give a better visualization of the results.
“This is because there are no significant gaps between the use of 3 to 12 features in terms of accuracy (Figure 5). The accuracy in this context refers to the overall probability of the defined three classes which is accurately estimated during the measurement, i.e. independent of the specific LOS, NLOS, and MP conditions. As a rule of thumb, fewer features in the model typically allow lower computation and better resource efficiency, especially for MCU-based platforms. In terms of the test time, the performance of MLP degraded when less than 5 features were applied. However, there were no notable differences in SVM and RF when more than 2 features were used in the evaluation (Figure 5).”
- Please define the accuracy in Fig. 5-6. Is the probability that the channel condition is accurately estimated (independently of the specific LOS/NLOS/MP)?
Thank you very much for mentioning the definition of accuracy in Fig. 5-6. Yes, it is the overall probability of the channel condition, which is accurately estimated independently of the specific classes LOS, NLOS, and MP conditions as you precisely mentioned. We added the following sentences in the revised manuscript as your suggestion.
“The accuracy in this context refers to the overall probability of the defined three classes which is accurately estimated during the measurement, i.e. independent of the specific LOS, NLOS, and MP conditions.”
- Can you please compare the ML-inspired approaches with others in the literature?
We would like to thank you for your suggested excellent idea of comparing the ML-inspired vs. traditional approaches in UWB. Indeed, a good review paper regarding this comparative analysis could be separately written based on your suggested opinion. As a matter of fact, a lot of papers regarding ML-inspired NLOS identification in UWB have been emerging. At the same time, there is also enough literature about traditional approaches in this matter. This is also one of the reasons why we separated the literature review section into two parts (ML-inspired and traditional approach). However, a good comparative analysis needs rigorous system evaluations for all the presented methods. We are sorry to say that we couldn’t do it in such a short period of time at the moment. As we mentioned above, we may write a review paper about this comparative analysis based on your suggested opinion in the future. Thank you very much again for directing us to this excellent idea.

Round 2
Reviewer 2 Report
After reading the authors' comments and their modifications to the text I think most of my initial doubts have been resolved. However, I think that some of the answers included in the response to the reviewer could also be included (obviously by adapting the text) in the final work, as they provide information that is not present in the original text. For example, the answer to question 4 is much richer in content than the final text that has been included in the revised version. I would recommend to authors that they consider expanding this part to make it clear that they have analyzed various alternatives when explaining the nature of the errors in the so-called MP scenario.
Author Response
Comments and Suggestions for Authors
After reading the authors' comments and their modifications to the text I think most of my initial doubts have been resolved. However, I think that some of the answers included in the response to the reviewer could also be included (obviously by adapting the text) in the final work, as they provide information that is not present in the original text. For example, the answer to question 4 is much richer in content than the final text that has been included in the revised version. I would recommend to authors that they consider expanding this part to make it clear that they have analyzed various alternatives when explaining the nature of the errors in the so-called MP scenario.
We would like to gratefully thank you for your positive feedback, constructive comments, and valuable time for reviewing this manuscript. We added the following sentences in the revised manuscript as your suggestion:
“The research data of the mentioned work is publicly available in [12].”
“Regarding the ranging errors in UWB at a static scenario, a more rigorous evaluation was conducted in our previous work [8], where its corresponding experimental research data were given in [12].”
“One of the reasons the error occurred in the MP condition at this scenario is because of the preamble accumulation time delay (PATD) in the coherent receiver of the UWB chip [8]. PATD is affected by the presence of multipath in wireless measurements. It is more notable when the arrivals of the reflected signal are within the chip period of the first path signal [8]. Moreover, it is worth mentioning that the error deviations in Figure 2(d) correspond to a single range (a Tag to an Anchor) in the measurement. In general, at least three ranges (usually more in multilateration methods) of such measures are necessary for UWB localization [6,14]. This implies the combination of such errors in a ranging phase contributes to a significant impact on the overall system performance.”
